# Multi-faceted epigenetic dysregulation of gene expression promotes esophageal squamous cell carcinoma

Wei Cao [1,15✉], Hayan Lee [2,15], Wei Wu [3,4,15✉], Aubhishek Zaman[3,4,15], Sean McCorkle[5], Ming Yan[6], Justin Chen[2], Qinghe Xing[7], Nasa Sinnott-Armstrong [2], Hongen Xu[8], M. Reza Sailani[2], Wenxue Tang[8], Yuanbo Cui[1], Jia liu[1], Hongyan Guan[1], Pengju Lv[1], Xiaoyan Sun[1], Lei Sun[1], Pengli Han[1], Yanan Lou[1], Jing Chang[9], Jinwu Wang[10], Yuchi Gao[11], Jiancheng Guo[8], Gundolf Schenk [12], Alan Hunter Shain[13], Fred G. Biddle[14], Eric Collisson[3,4], Michael Snyder [2✉] & Trever G. Bivona [3,4✉]

Epigenetic landscapes can shape physiologic and disease phenotypes. We used integrative, high resolution multi-omics methods to delineate the methylome landscape and characterize the oncogenic drivers of esophageal squamous cell carcinoma (ESCC). We found 98% of CpGs are hypomethylated across the ESCC genome. Hypo-methylated regions are enriched in areas with heterochromatin binding markers (H3K9me3, H3K27me3), while hyper-methylated regions are enriched in polycomb repressive complex (EZH2/SUZ12) recognizing regions. Altered methylation in promoters, enhancers, and gene bodies, as well as in poly-comb repressive complex occupancy and CTCF binding sites are associated with cancer-specific gene dysregulation. Epigenetic-mediated activation of non-canonical WNT/β-catenin/MMP signaling and a YY1/lncRNA ESCCAL-1/ribosomal protein network are uncovered and validated as potential novel ESCC driver alterations. This study advances our understanding of how epigenetic landscapes shape cancer pathogenesis and provides a resource for biomarker and target discovery.

[1] Translational Medical Center, Zhengzhou Central Hospital Affiliated Zhengzhou University, Zhengzhou, China. [2] Department of Genetics, School of Medicine, Stanford University, CA, USA. [3] Department of Medicine, University of California San Francisco, San Francisco, CA, USA. [4] Helen Diller Family Comprehensive Cancer Center, University of California San Francisco, San Francisco, CA, USA. [5] Computational Science Initiative, Brookhaven National Laboratory, Upton, NY, USA. [6] Basic Medical College, Zhengzhou University, Zhengzhou, China. [7] Institutes of Biomedical Sciences and Children's Hospital, Fudan University, Shanghai, China. [8] Precision Medicine Center, The Academy of Medical Sciences, Zhengzhou University, Zhengzhou, China. [9] Jiangsu Mai Jian Biotechnology Development Company, Wuxi, China. [10] Department of Pathology, Linzhou Cancer Hospital, Linzhou, China. [11] Annoroad Gene Company, Beijing, China. [12] Bakar Computational Health Sciences Institute, University of California San Francisco, San Francisco, CA, USA. [13] Department of Dermatology, University of California San Francisco, San Francisco, CA, USA. [14] Department of Biological Sciences, University of Calgary, Calgary, Canada. [15]These authors contributed equally: Wei Cao, Hayan Lee, Wei Wu, Aubhishek Zaman. ✉email: caowei7@zzu.edu.cn; wei.wu@ucsf.edu; mpsnyder@stanford.edu; trever.bivona@ucsf.edu

Epigenetic regulation is an important determinant of many biological phenotypes in both physiologic and pathophysiological contexts[1]. Nevertheless, epigenetic forces that shape the evolution of complex diseases, such as cancer, remain incompletely defined. Esophageal cancer is the sixth leading cause of cancer-related death and the eighth most common cancer worldwide[2]. In China and East Asia, esophageal squamous cell carcinoma (ESCC) is the most prevalent pathohistological type of esophageal cancer[3]. Comprehensive analysis by whole-genome and whole-exome sequencing uncovered the genetic landscape of ESCC[4–9] and multi-region whole-exome sequencing revealed intra-tumor genetic heterogeneity in ESCC[10]. This intra-tumor genomic heterogeneity could serve as a prognostic predictor in esophageal cancer[11] and as a foundation for improved treatments. Notable and frequently mutated epigenetic modulator genes in ESCC include KMT2D, KMT2C, KDM6A, EP300, and CREBBP, and epigenetic perturbations might interact with other somatic genomic alterations to promote the progression of ESCC. The interplay between epigenetic perturbations and other somatic genetic alterations may play a critical role during ESCC tumorigenesis[4].

The Cancer Genome Atlas research group (TCGA) identified ESCC-related biomarkers at different multi-omics levels (genomic, epigenomic, transcriptomic, and proteomic) and highlighted 82 altered DNA methylation events, along with transcriptional and genomic alterations[9]. While genomic and transcriptomic-level studies of ESCC produced valuable biological discoveries and resources, the single-nucleotide resolution of the epigenetic landscape of ESCC, and of most other cancers, at the whole-genome level remains poorly studied. This knowledge gap is due to the comparatively high cost, computational complexity, and technical challenges of capturing genome-wide single-nucleotide resolution of the epigenetic landscape. Integrative and causal analyses using orthogonal multi-omics datasets are incomplete without a high-resolution methylome profile. We addressed this challenge by using an integrated multi-omics study that includes whole-genome bisulfite sequencing (WGBS), whole-genome sequencing (WGS), whole-transcriptome sequencing (RNA-seq), and proteomic experiments on a cohort of ESCC samples and their adjacent non-tumor esophageal tissues, along with orthogonal analysis and validation using the large TCGA-esophageal cancer (ESCA) dataset. Our goal was to understand the extent and complexity of DNA methylation alterations and their consequent dysregulation of both protein-coding and non-coding gene expression.

## Results

### WGBS reveals the epigenetic landscape in ESCC.
Different types of cancers exhibit unique epigenetic alterations, particularly in the DNA methylome[12–14]. We initially collected ten pairs of primary ESCC samples and their adjacent non-tumor tissues (Supplementary Fig. 1), performed WGBS with over 99% of a bisulfite conversion ratio, and generated a mean 15× sequencing depth per sample (Supplementary Data File 1, Supplementary Fig. 2). Over 99% of CpG dinucleotides were covered and ~95% of CpGs were reliably mapped by more than five reads. We orthogonally validated approximately 300K CpG methylation changes detected in our sample cohort with their methylation level changes detected in TCGA-ESCA sample cohort using Human Methylation 450K (HM450K) array (Supplementary Fig. 3a) and showed overall high correlation of CpG methylation changes in all normal and tumor samples between the two cohorts (Pearson's r = 0.9644, p value < 0.01, Supplementary Fig. 3b), and within subtype of tumors (Pearson's r = 0.7570 for our ESCC versus TCGA-ESCC, Pearson's r = 0.5554 for our ESCC versus TCGA-EAC

(esophageal adenocarcinoma), Supplementary Fig. 3c, d). DNA methylation at non-CpG contexts was present in less than 0.5% in our samples.

More than five million significantly differentially methylated cytosines (DMCs) were identified (one-way ANOVA test, multiple hypothesis tests were adjusted by the Benjamini–Hochberg method, q value < 0.05) (Fig. 1a). Among them, 57.5% of DMCs were located at known annotated regions (e.g., introns, exons, promoter and enhancer regions, and CpG islands) and 42.5% were located at unannotated regions of the genome (Supplementary Fig. 4a). Methylation loss in cytosines in ESCC accounted for 97.3% of the DMCs and was mostly confined to intergenic regions of the genome. Only 2.7% of the DMCs were gains of methylation in ESCC (proportional test for hyper- and hypo-methylation, p value < 2.2e−16, Fig. 1b) and 83.67% of them mapped to gene bodies, promoters and enhancers, and CpG islands with RefSeq annotation (Supplementary Fig. 4b, c). Of the hypomethylated DMCs in ESCC, 63.08% were mapped to lncRNA regions with ENCODE annotation (v27lift37), which is significantly higher than that in random regions (permutation test, p = 0.001, Z-score = 15.021, 1000 permutation), whereas 58.01% of hypermethylated DMCs in ESCC were dispersed in antisense RNA regions of the genome (Supplementary Fig. 5a, b). We found 36.24% and 69.79% of DMCs overlap with enhancers and CpG islands, respectively (permutation test, p value = 0.00099, number of iterations = 1000), which are dispersed in regulatory regions in the genome.

### Epigenetic features shape cellular identity.
We found that the genome-wide CpG methylation levels can discriminate normal tissues from tumor tissues in an unbiased manner, as measured by unsupervised principal component analysis (PCA) (Fig. 1c), similar to unsupervised transcriptome-mediated clustering of normal and tumor samples (PCA in Supplementary Fig. 6a and Dendrogram in Supplementary Fig. 6b). In the larger sample set (n = 202) of TCGA-ESCA, CpG probes present in the lower resolution Illumina HM450K array were also able to discriminate normal and tumor and even subtypes of esophageal cancer using t-distributed stochastic neighbor embedding (t-SNE), a nonlinear dimensionality reduction algorithms (Fig. 1d). Twenty-seven of DMC probes were identified in ESCC versus EAC using the criteria of absolute log2(FC) ≥ 0.59 and FDR < 0.05 (Supplementary Fig. 6c). The data suggest that alterations in DNA methylation can characterize the distinct biological features of cellular states of physiologic and pathophysiologic phenotypes. DNA methylation heterogeneity has been observed in other cancer types[14,15] and stochastically increasing variation in DNA methylation appears to be a property of the cancer epigenome[16]. We found significantly higher variance of methylation changes in ESCC (275.76 ± 204.01) compared with normal samples (95.67 ± 112.38, two-sample t-test, p value ≈ 0) in our cohort. This was also observed in our analysis of TCGA-ESCC cohort (n = 98, two-sample t-test, p < 2.2e−16) (Supplementary Fig. 6d). As a further measurement of the level of epigenetic variance, we calculated Shannon's entropy of methylation levels at each CpG locus. We observed increased entropy in ESCC compared with normal samples (two-sample t-test p value ≈ 0) (Fig. 1e). This is consistent with the increase in stochastic noise (heterogeneity) in tumors. Our simulation using the Euler–Murayama method[17] also reflected increased DNA methylation heterogeneity in ESCC (Supplementary Fig. 6e).

The clinical significance of such high variance of DNA methylation changes in cancer remains unclear. Using the independent TCGA-ESCC clinical cohort, we stratified patient samples into low or high variance groups by their median

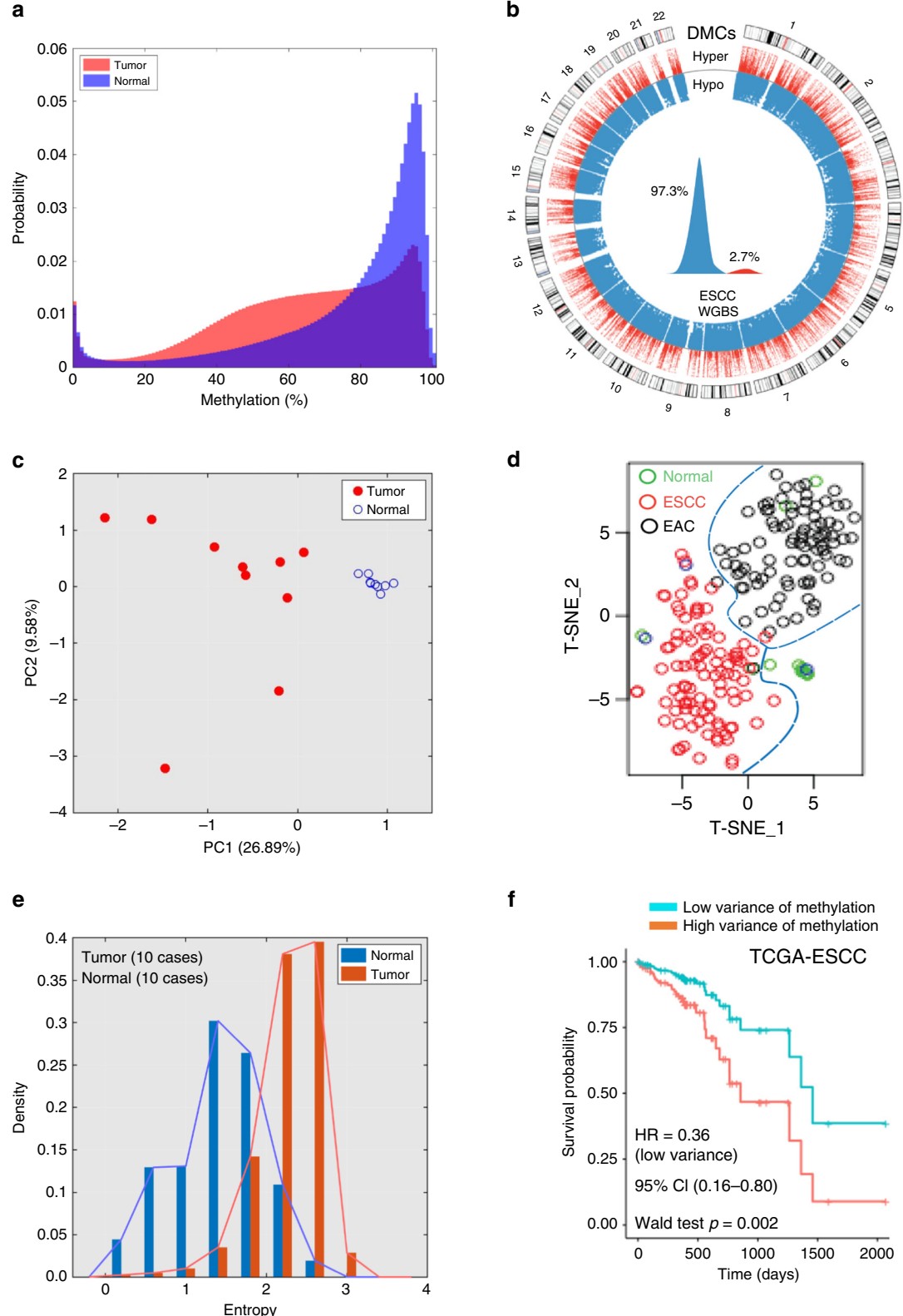

variance of methylation level along with other clinical variables (age, gender, alcohol usage) for multivariate Cox regression analysis. Although heavy alcohol intake is a known risk factor in ESCC development[18], we observed a trend toward to inferior overall survival time in patients with alcohol consumption but no impact on methylation variance: only three DMC probes associated with alcohol users ($\log_2$(FC) $\geq 0.2$ and FDR $< 0.05$)

(Supplementary Fig. 7a–c). The group with a lower variance ($n = 49$) of methylation levels showed a favorable overall survival time (hazard ratio $= 0.36$, 95% confidence interval (0.16–0.80), Wald test $p$ value $= 0.002$) after normalized to age, gender, and alcohol consumption (Fig. 1f). We examined additional squamous types of cancer including TCGA-head and neck squamous carcinoma cohort ($n = 516$) and found that again the high variance of DNA

**Fig. 1 Epigenetic landscape and heterogeneity in esophageal squamous cell carcinoma (ESCC). a** Ten pairs of ESCC and adjacent normal tissues were performed whole-genome bisulfite sequencing (WGBS). The asymmetric density distribution of all CpG methylation statuses in the normal esophageal tissues versus ESCC. ESCCs lose methylation which leaves most CpGs partially methylated. Normal = blue, tumor = red. **b** Circos plot of >5 million differentially methylated CpGs (DMCs) between ESCC tumor and adjacent normal tissue. DMCs are substantially hypomethylated in ESCC (97.3%). Only 2.7% are hypermethylated in ESCC. **c** Principal component analysis (PCA) shows that characteristic CpGs discriminate tumor samples from normal samples. **d** t-Distributed Stochastic Neighbor Embedding (t-SNE) showed CpG methylation profiling of TCGA-esophageal cancer from human methylation 450K analysis clustered into either normal tissue (n = 15, green circles) or ESCC (n = 97, red circles) or esophageal adenocarcinoma (n = 89, black circles) subtypes. **e** Entropy analysis of all CpGs showed variations per CpG in normal esophageal tissues (blue bars) and ESCC (red bars). The entropy of CpGs in ESCC was higher than in normal samples. **f** Multivariate cox proportional hazard analysis demonstrated TCGA-ESCC patients (n = 92) with lower variance of CpG methylation in tumors showed better survival time than those with higher variance. Median variance was used to discriminate high versus low variance groups. Variance of DNA methylation changes were normalized for age, gender, and alcohol consumption in patients. Statistical significance was assessed by two-sided Wald test, $p = 0.002$.

methylation changes associates with poor survival outcome ($p$ value $<2e-16$) (Supplementary Fig. 7d). The above findings provide potential clinical relevance of the epigenetic heterogeneity within ESCC.

**Differentially methylated regions (DMRs) associate with chromatin modifications.** We further defined 295,295 DMRs ($p$ value $\leq 0.05$, FDR $\leq 0.05$) between tumor and normal tissues, resulting from a CpG density peak of 4% and a DMR peak size of 200–400 base pairs (bp) (Fig. 2a, Supplementary Fig. 8a, b). Only 1.8% of these DMRs are hypermethylated, while 98.2% of DMRs are hypomethylated (proportional test, $p$ value $<2.2e-16$) in tumors relative to normal tissues. DMRs in regions of $-2990$ to $+6990$ bp to transcription start site (TSS) appear hypermethylated while gene bodies, intergenic, and non-coding regions are in general hypomethylated in tumors (Fig. 2b). The DMRs distribution in Chromosome (Chr)8, Chr19, and Chr20 is higher than in other Chrs after normalization to chromosome size (Supplementary Fig. 9a) and DMRs are enriched in gene promoters at Chr19 ((Supplementary Fig. 9b (normalized by gene size), 9c (normalized by gene counts in each Chrs)).

We integrated the most significant DMRs with all CpGs, CpG island, chromatin state, and potential transcription factor (TF) binding data using the ENCODE dataset[19]. We observed Chr8 harbors three large genomic regions with unique DMR patterns and these regions contain the *SOX17*, *RGS22*, and *ESCCAL-1* (*CASC9*) gene loci, respectively. For example, around the gene of *SOX17* (Chr8: 55,360,000–55,400,000), CpG island regions were hypermethylated but CpG shore regions were hypomethylated (Fig. 2c). Two CpG islands with a significant hypermethylation upstream of the *SOX17* gene were observed but that was not associated with low gene expression ($p$ value $= 0.668$, Fig. 2d). In the region 100,650,000–101,190,000 of Chr8, hypo-DMRs covered all of the gene body of *RGS22* (regulator of G protein signaling), which is a putative tumor suppressor[20]. This was associated with decreased RGS22 expression ($p$ value $= 0.2$) (Supplementary Fig. 10). The region around the lncRNA ESCCAL-1 (Chr8:76,130,000–76,240,000), which was previously identified by us[21], contained significantly hypomethylated DMRs in its promoters and we further investigated the uncharacterized biological function of this lncRNA later in this study.

TFs play important biological roles in gene regulations and their binding affinities can be affected by DNA methylation changes[22]. The occupancy of each TF binding consensus varies in the genome (161 TF-binding sites from ENCODE), with POLR2A (DNA-directed RNA polymerase II subunit A) (5.23%) and CTCF (CCCTC-binding factor) (3.55%) ranking at the top (Supplementary Fig. 11a, Supplementary Data File 2). We searched CpG content in these TF-binding sequences and the top 20 TFs affected by methylation alterations in consensus-binding sites were identified. Notably, the Polycomb Repressor Complex 2

(PRC2) subunits SUZ12 and EZH2 binding sites were substantially affected by hypermethylation in the CpGs (Supplementary Fig. 11b–d). EZH2 and SUZ12 and other TFs tend to preferentially bind to promoter regions compared to enhancer regions ($p$ value $<0.001$) (Supplementary Fig. 12a–c). These observations indicated the possibility of a paradoxical activation mechanism for PRC2 target genes through loss of PRC2 occupancy in gene promoters in tumor cells.

A link between hypomethylated blocks, variable gene expression, and large heterochromatin domains such as Large Organized Chromatin lysine (K) modification (LOCK) or lamina-associated domains was previously reported in cancer[23]. We performed the genomic region set enrichment analysis with LOLA[24] to test these DMRs for enrichment against the LOLA Core database, which contains DNaseI hypersensitive elements and chromatin immunoprecipitation sequencing (ChIP-seq) peaks from a variety TFs and chromatin modifiers. ESCC-derived hypo-DMRs were enriched in genomic regions with heterochromatin markers such as H3K9me3 and H3K27me3, whereas hyper-DMRs were enriched in genomic regions with EZH2 or SUZ12 protein-binding sites (Fig. 2e, Supplementary Data File 3). The DMRs enriched in regulatory elements in cancer indicates cancer-specific dysregulation of gene expression contributing to cancer progression, which we explore further later in this study.

**Aberrant DNA methylation in promoter regions mediates transcriptional dysregulation in ESCC.** From our WGBS analysis, we identified 4391 promoter regions ($-4500$ to $+500$ bp to TSS) of coding and non-coding genes whose CpGs were significantly differentially methylated (FDR $< 0.01$). Functional annotation of these target genes harboring promoter hypomethylation indicated an over-representation of WNT/β-catenin signaling, whereas gene promoters harboring hypermethylation were enriched for KIT signaling genes (Supplementary Fig. 13a). The RNA-seq dataset displayed 4074 significantly differentially expressed genes (DEGs) in ESCC relative to the adjacent normal tissues. The functional annotation of these DEGs indicated enrichment for genes regulating cell cycle pathways and metallopeptidase activity (Supplementary Fig. 13b).

DNA methylation at regulatory regions can influence transcript expression levels[25]. We merged DMRs and DEGs and identified 694 genes that showed significant differential methylation alteration in promoters and concomitant dysregulation of gene expression (Fig. 3a). The genes were systematically classified into four distinct clusters (denoted as C1, C2, C3, and C4) according to methylation and gene expression pattern (Fig. 3b). Genes in C1 and C2 followed the well-documented canonical model, showing anti-correlation in promotor methylation and gene expression[13]; in contrast, genes in C3 and C4 showed a non-canonical pattern in that promotor methylation and gene

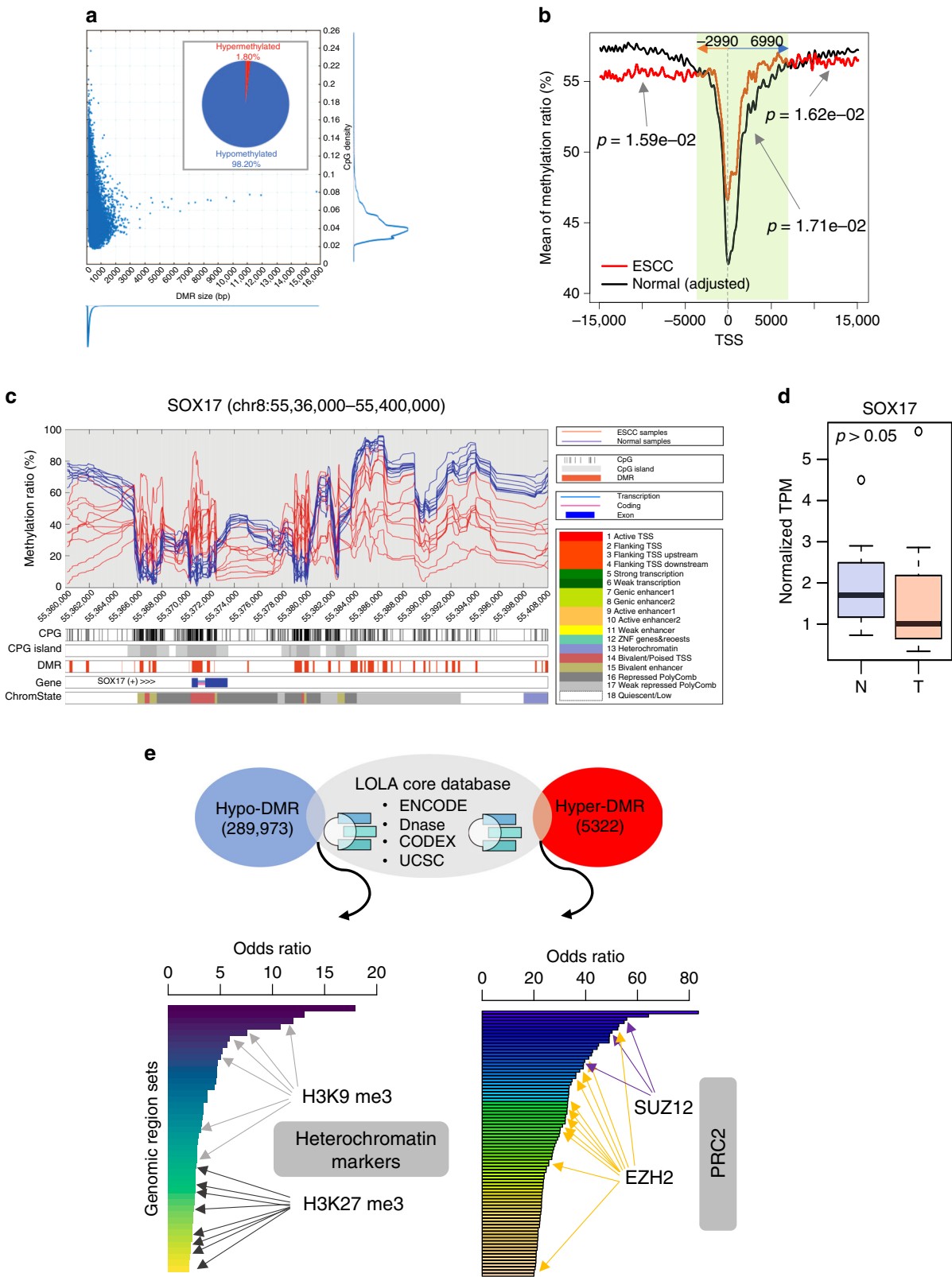

expression was positively correlated (Fig. 3c, Supplementary Data File 4). Among the 694 genes, only 1.5% of them harbor non-synonymous mutations from our selected cases of WGS and no copy number changes of these genes were detected from WGS and inferred from RNA-seq analysis (Supplementary Fig. 15). As TF expression can lead to changes in gene expression, we examined the RNA expression of 161 TFs in our ESCC samples and found no significant changes in expression of these TFs in tumors versus normal samples (Supplementary Fig. 16). There-fore, the majority (98.5%) of these dysregulated genes in cluster C1–C4 appear to occur via epigenetic dysregulation (epimuta-tion)[26]. This phenomenon was recapitulated in the independent TCGA-ESCC (n = 96) sample cohort with available multi-OMICs datasets (Supplementary Fig. 17a–d).

**Fig. 2 Differentially methylated regions (DMRs) and their functional impacts on the ESCC genome. a** A DMR identification algorithm from DMC was developed using two criteria: (1) two flanking DMCs should be close (150 base pairs, bp) given the minimum size of CpG island (CGI) 150 bp and (2) the methylation pattern should be consistent, either hypomethylated or hypermethylated within a DMR. Our algorithm revealed the distribution of DMR size and CpG density within DMRs. Both distributions of DMR size and CpG density are asymmetric and have long tails as DMR size increases in length and CpG density is more compact. The peak of DMR size is 200–300 bp and the peak of CpG density is approximately 4% (also seen in Supplementary Fig. 8). **b** Methylation level of CpGs within 15,000 bp upstream and downstream relative to a transcription start site (TSS) was assessed in ESCC and normal esophagus separately. Overall methylation conversions between normal esophagus tissue and ESCC were observed. CpGs in ESCC tend to be hypermethylated between 2990 bp upstream and 6990 bp downstream of a given TSS. The arrow indicates the *p* value for the specific region of significant methylation changes. **c** A representative genomic region at chr8:55,360,000–55,400,000 with hypermethylation in CpG island and hypomethylation in the CpG shore. **d** SOX17 expression is decreased in ESCC tumors (T, 10 cases) relative to normal tissue (N, 10 cases). Y-axis is the normalized gene expression levels (transcript per million reads, TPM) from RNAseq data. Box and whisker plot: center line, median; box limits, upper and lower quartiles; and whiskers, maximum and minimum values, circle dots, outliers. Statistical significance was assessed by two-sided *t*-test, $p = 0.67$, FDR = 0.8. **e** Genomic regional enrichment analysis of ESCC-associated DMRs. In all, 289,973 hypo-DMRs and 5322 hyper-DMRs were mapped to LOLA core database with ENCODE, DNase, CODEX, and UCSC genomic annotations. The significant overlapping genomic regions were selected with *p* value <0.05 and odds ratio >2 from Fisher's exact test. Each bar with pseudocolor gradience represents each dataset in LOLA core databases (see Supplementary Data File 3).

The underlying mechanisms of the divergent regulation of gene expression are complex and involve DNA methylation, chromatin remodeling, and DNA accessibility[27]. We explored a potential explanation for the non-canonical patterns in C3 and C4 across epigenetic regulatory features. First, we cross-referenced 13,000 TCGA-ESCA chromatin-accessible regions as determined by ATAC-seq[27] to the regulatory regions of the 694 genes. Accessibility in promoter regions was higher in C2 and C3 than in C4 and C1, respectively (Supplementary Fig. 18a, b). Second, methylation in defined promoter regions and in gene bodies showed a differential phenotype between C1 and C3 (but not between C2 and C4): methylation levels in gene bodies were higher in C3 ($-5.4587 \pm 26.3450$) than in C1 ($-26.8551 \pm 16.4716$, $p < 0.05$). There is no significant difference in related enhancer regions of genes in all clusters (Fig. 3d). Third, hypermethylation at cohesion and CCCTC-binding factor-binding sites could compromise binding of this methylation-sensitive insulator protein and result in gene activation[22]. Thus, we searched for CTCF-binding sites within promoter regions of the 694 genes and observed that the CTCF-binding sites were enriched in C3 (Fig. 3e), which could partially explain the phenotype of high promoter methylation and high gene expression. The data also indicated that the promoters of genes in C4, despite being hypomethylated in the tumor, have scarce accessible regions (Supplementary Fig. 18b). This highlights the importance of both accessibility and absence of methylation as linked features of the gene expression pattern in the C4. Enhancer–promoter interaction is a fundamental mechanism of gene regulation[28]. We found the pattern of enhancer-gene expression was highly correlated to promoter-gene expression pattern, with the exception of non-canonical C3 and C4 groups, in which the methylation change in enhancers was opposite to that in promoters (Supplementary Fig. 14b–d).

Functional annotation of the 694 genes was performed using multiple databases (KEGG[29], WikiPathways[30], ENCODE[19], ChEA[31]) and showed that PRC2 subunit (EZH2 and SUZ12)-mediated polycomb repressive gene sets were enriched in the non-canonical clusters C3 and C4 (Fig. 4a). We searched for ENCODE-defined EZH2 and SUZ12-binding sites across gene promoters in C1–C4 and observed that EZH2 occupancy was enriched in C3 ($1.5970 \pm 1.2316$) and C4 ($0.6000 \pm 0.7684$) compared with C1 ($0.9167 \pm 0.8464$) and C2 ($0.2336 \pm 0.5870$), respectively (*p* value < 0.001) (Fig. 4b). SUZ12 occupancy was higher only in C3 gene promoter regions ($1.5522 \pm 1.7946$) (Fig. 4c). To understand the functional mechanism that is responsible for differential methylation at target-gene promoters, we performed unsupervised hierarchical clustering of ENCODE-defined known TF-binding sites in 694 gene promoters.

The analysis showed that EZH2 and SUZ12 binding sites clustered together and were enriched in genes in C3 compared to other TFs in other clusters (Fig. 4d, Supplementary Figs. 19 and 20). Increased WNT2 gene expression was significantly associated with increased methylation in promoter regions in C3 (Fig. 4e). The comprehensive analyses show the non-canonical gene expression pattern (C3) appears to arise via de-repression of the EZH2-mediated suppressor effects on promoter regions of genes in C3 to increase gene expression, which we experimentally validate later in this study.

**DNA methylation gain at the promoter region activates the WNT2/ β-catenin pathway in ESCC.** WNT2 belongs to the structurally related WNT family of genes that function as secretory ligands for the WNT signaling pathway[32]. Canonical WNT signaling pathway results in stabilization of the transcriptional co-regulator β-catenin and subsequent upregulation of downstream target genes[32].

Epigenetic dysregulation of the components of MAPK, AKT, and WNT pathway can promote aberrant activation of these pro-growth pathways in ESCC[33]. In our RNAseq dataset, we identified only WNT2 in the WNT pathway as significantly highly expressed in the tumor samples compared to normal samples (Supplementary Fig. 21a, b). WNT2 protein was highly expressed in the cytoplasm of cancer cells relative to normal cells (Supplementary Fig. 22a, b). Although high expression of WNT2 is not a prognostic marker, it was associated with tumor progression (Supplementary Fig. 22c). These results indicate selective and specific upregulation of WNT2 in ESCC tumors through a putative non-canonical epigenetic regulatory mechanism.

To gain mechanistic insight into the epigenetic regulation of the WNT2 promoter, we queried our TF target-gene hierarchical clustering analysis for genes in C3 and found that the EZH2-binding site along with SUZ12-binding sites were present within the WNT2 promoter region compared to other regulatory factors (Fig. 4d, Supplementary Fig. 19b). EZH2 and SUZ12 are subunits of PRC2, which has histone methyltransferase activity to primarily tri-methylate histone H3 on lysine 27 (H3K27me3)[34] resulting in gene silencing. We also found that the EZH2-binding site within the WNT2 promoter overlaps with the hyper-CpG methylation sites in the WNT2 promoter region in cancer cells (Fig. 5a).

The WNT2 promoter region (Chr7: 116,960,000–116,965,000) was hypermethylated, but paradoxically associated with increased gene expression in tumors (Supplementary Fig. 23 from our WGBS dataset, Supplementary Fig. 24a–c from TCGA-dataset). The WNT2 gene shows rare mutation (0.55%, 1/183 cases) and

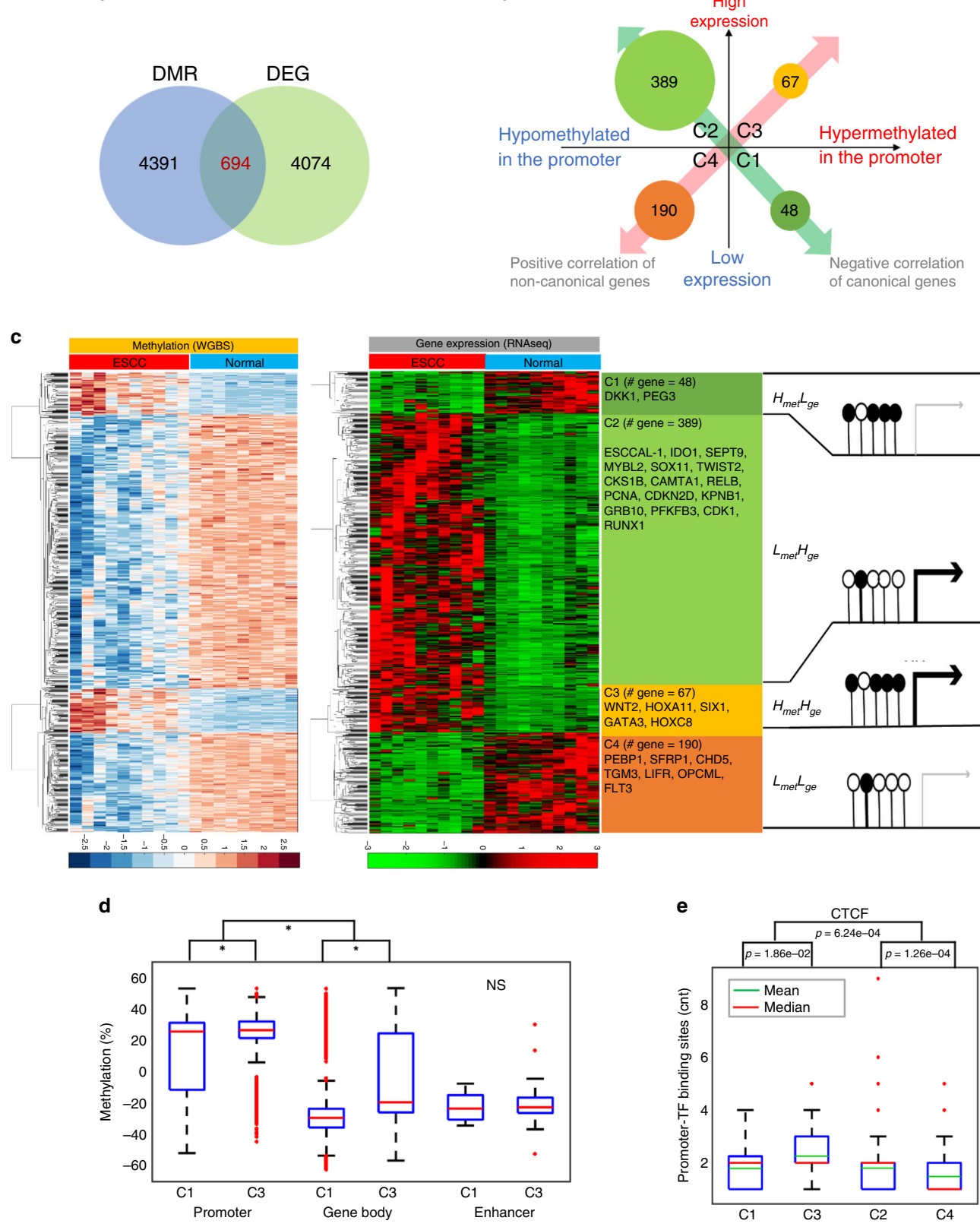

copy number alteration (16%, 30/183 shallow loss; 4.4%, 8/183 shallow gain) in TCGA-ESCA cohort (Supplementary Fig. 24d). Thus, we reasoned that de-repression of EZH2 occupancy may cause non-canonical methylation-mediated activation of WNT2 gene expression in ESCC. We validated EZH2 occupancy on the DMRs of the *WNT2* promoter by performing chromatin immunoprecipitation sequencing (ChIP-seq) in normal immortalized esophageal epithelial cells (Het-1A) and the patient-derived ESCC cell line, EC109. The ChIP-seq analysis showed EZH2-binding peaks at the *WNT2* promoter region in normal cells compared to minimal binding peaks in the ESCC cells (Fig. 5a). Furthermore, we confirmed the promoter

**Fig. 3 Integrative analysis of WGBS and RNAseq uncovered methylation-mediated diverse gene regulation. a** A total of 694 genes were selected for the methylome–transcriptome association analysis. The selected genes have statistically significant DMRs (FDR ≤ 0.001) in promoters, defined as 4500 base pair (bp) upstream and 500 bp downstream relative to transcription start sites, and are statistically significant DEGs (differentially expressed genes) (|log2 (fold change)| > 1, FDR ≤ 0.05). **b**, **c** The association of promoter methylation and expression of the 694 genes were identified. There are four clusters: C1 ($n = 48$): genes that are hypermethylated in the promoter with low expression level in ESCC; C2 ($n = 389$): genes that are hypomethylated in the promoter with high expression in ESCC; C3 ($n = 67$): genes that are hypermethylated in the promoter with high expression in ESCC; C4 ($n = 190$): genes that are hypomethylated in the promoter with low expression in ESCC. Genes in C1 and C2 fit the canonical model of regulation, while genes in C3 and C4 are not well explained by current understanding. Representative genes are listed in each cluster. See Supplementary Data File 4. **d** The quantification of CpG methylation in gene promoters and gene bodies in C3 ($n = 67$) is significantly higher than in C1 ($n = 48$), $p < 0.001$, while no significant difference (NS) for enhancers. **e** CTCF-binding sites are significantly higher in C3, indicating hypermethylation of inhibitors leads to de-repression to promote gene expression. The sample sizes for C1, C2, C3, and C4 in **d** and **e** are the same as defined **b** and **c**. Box and whisker plot: center line, median or mean; box limits, upper and lower quartiles; and whiskers, maximum and minimum values, circle dots, outliers. Statistic significance was assessed by one-way ANOVA.

region of *WNT2* was hypermethylated in three esophageal cancer cell lines (EC9706, EC109, and EC1) while no methylation was detected in normal esophagus epithelial cells (Het-1A) (Fig. 5b). Accordingly, WNT2 protein expression was higher in three cancer cell lines (Fig. 5c). DNA methyltransferase inhibitor (5-Azacytidine) treatment partially decreased the methylation level in the promoter region of WNT2 (Fig. 5d) and increased EZH2 binding (Fig. 5e), leading to the attenuation of WNT2 expression in cancer cells (Fig. 5f). In tissue samples, WNT2 mRNA and protein expression was also increased (Fig. 5g, h).

To identify downstream effector genes of WNT/β-catenin signaling that might promote ESCC, functional annotation of differential expression from proteomic data and Gene Set Enrichment Analysis (GSEA) from RNAseq data revealed extracellular matrix organization (Supplementary Fig. 25) and extracellular metalloproteins (MMP3 and MMP9, known β-catenin targets)[35] gene sets (Supplementary Fig. 26) were enriched in tumor samples. We validated that both MMP3 and MMP9 transcripts and proteins were highly expressed in ESCC relative to normal tissues (Fig. 5h, Supplementary Fig. 26a). As WNT2 RNA expression in ESCC cell lines is significantly higher than in normal cell lines (Supplementary Fig. 27a), we tested whether WNT2-mediated signaling was required for tumor cell growth, we suppressed WNT2 expression using two independent short interfering (si)RNAs in two ESCC cell lines (EC9706 and EC109) (Supplementary Fig. 27b, c). WNT2 knockdown significantly inhibited ESCC cell growth ($p$ value < 0.01) (Supplementary Fig. 27d, e). WNT2 knockdown decreased expression of MMP3 and MMP9, known targets of WNT/β-catenin signaling (Fig. 5i). Since MMPs can promote tumor invasion and metastasis[36], we tested whether WNT2 knockdown abrogates the migratory and invasive potential of ESCC tumor cells. In two ESCC cell lines (EC9706 in Fig. 5j, and EC109 in Supplementary Fig. 27f, g), silencing of WNT2 significantly reduced cellular invasion and migration ($p$ value < 0.01). The protein but not RNA levels of β-catenin, encoded by *CTNNB1* gene, was significantly elevated in ESCC tumors, which indicates that β-catenin protein has higher stability in these tumors (Supplementary Fig. 27h). Knockdown of WNT2 reduced the elevated protein level of β-catenin in patient-derived cell lines, as expected by previous findings[37,38] (Supplementary Fig. 27i). Moreover, WNT2 knockdown in EC9706 tumors significantly suppressed tumor growth and tumor burden (Student's *t*-test, $p$ value < 0.05, Fig. 5k and Supplementary Fig. 27j). The combined in vitro and in vivo evidence showed that a WNT2/β-catenin/MMP signaling axis was required for tumor cell growth, migration, and invasion in ESCC. Together, our results demonstrate a non-canonical mechanism for increased WNT2 expression in the absence of EZH2-PRC2 occupancy of the WNT2 promoter with hypermethylated CpGs. This non-canonical epigenetic activation of

WNT2-mediated signaling and exerted functional consequences in ESCC progression (Fig. 5l).

**Epigenetic activation of lncRNA ESCCAL-1 is an oncogenic driver in ESCC**. Increasing evidence indicates dysregulation of lncRNAs during cancer progression and metastasis; however, the mechanisms of dysregulation and of action of lncRNAs in cancer are relatively poorly understood[39]. Previously, we showed that the lncRNA ESCCAL-1 was overexpressed in ESCC[21], and overexpression of ESCCAL-1 has been reported in other cancer types[40–43]. The mechanism underlying ESCCAL-1 upregulation in cancer is unknown.

We found ESCCAL-1 is one of the most notable candidates for increased gene expression in C2 in association with decreased methylation (Fig. 6a). One DMR in the promoter of ESCCAL-1 showed decreased CpG methylation in cancer, leading to increased transcription of lncRNA ESCCAL-1 (Fig. 6b, c). There was no mutation or copy number variation of ESCCAL-1 reported or observed in TCGA-ESCA genomic dataset or in our WGS data of ESCC patients (Fig. 6b, top panel, Supplementary Fig. 24d). Independent verification of the methylation status of the ESCCAL-1 promoter region showed 62.5% (20/32) hypomethylation in ESCC tumors versus 71.8% (23/32) hypermethylation in adjacent normal tissues (chi square test $p$ value < 0.01) (Fig. 6d, e). In agreement, ESCCAL-1 expression was significantly higher in ESCC compared to adjacent normal tissues ($p$ value = 0.00113, FDR < 0.05) (Fig. 6c, from RNAseq). We corroborated these observations by analysis of an independent cohort of 73 ESCC tissues relative to their normal counterparts (Fig. 6f). We also noted a hypermethylated ESCCAL-1 promoter region in normal esophageal cells (Het-1A), whereas methylation was not detected in three ESCC cell lines (EC1, EC109, and EC9706) (Fig. 7a). ESCCAL-1 expression was substantially overexpressed in ESCC cell lines compared with normal cells Het-1A (Fig. 7b). Furthermore, increased expression of ESCCAL-1 was a biomarker of worse overall survival time and progression-free survival in ESCC patients (Fig. 7c, d). Knockdown of ESCCAL-1 reduced growth of patient-derived ESCC cells in vitro (Supplementary Fig. 28) and in vivo (Fig. 7e, f), suggesting a cancer-promoting function.

To identify a possible mechanism of ESCCAL-1 upregulation, we examined sequence motifs of known TFs in the ESCCAL-1 hypomethylated promoter region in silico and found a predicted binding site for YY1. YY1 is an TF belonging to the GLI-Kruppel class of zinc-finger proteins and contributes to tumorigenesis[44]. Using ChIP-PCR, we validated YY1 binding at the hypomethylated promoter region of ESCCAL-1 (Fig. 7g). RNA-seq profiling in cells with YY1 knockdown also revealed ESCCAL-1 as a YY1-regulated target-gene (Supplementary Fig. 29). RT-PCR assays also showed decreased expression of ESCCAL-1 in cells with YY1

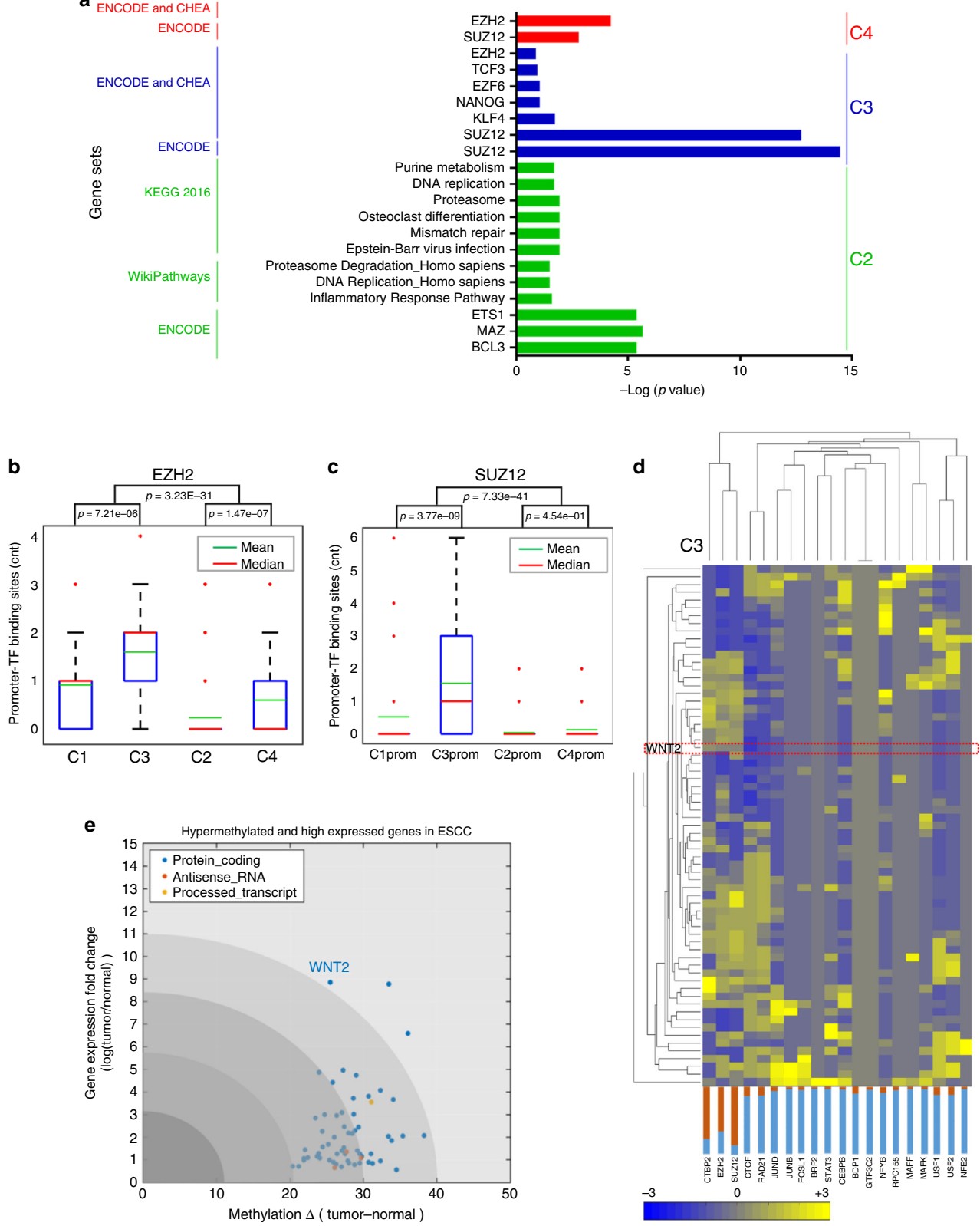

deficiency (Fig. 7h), indicating YY1 transcriptionally regulates ESCCAL-1 in ESCC.

Since the downstream mechanism of ESCCAL-1's contribution to ESCC pathogenesis was unclear, we performed a "guilty-by-association" co-expression analysis using our RNA-seq dataset. The ESCCAL-1-related gene expression modules are enriched in cell cycle pathways, RNA binding and the Myc pathway (Supplementary Fig. 30a–c). In order to explore the causative roles of ESCCAL-1 in ESCC progression, we conducted RNA-seq profiling in EC9706 cells with ESCCAL-1 knockdown by shRNA (or with shControl). We hypothesized that depletion of ESCCAL-1 could reverse the phenotype of cells to the relative normal cell

**Fig. 4 DNA methylation at regulatory consensus protein-binding sites and impacts on gene expression. a** Functional annotation for the four distinct methylation-transcriptome clusters. Lists of genes in C2 ($n = 389$), C3 ($n = 67$), and C4 ($n = 190$) were subjected to gene set analysis using hypergeometric statistics for gene sets collected from multiple databases (ENCODE, CHEA, KEGG, WikiPathways, Reactome, GO molecular function, Panther, BIOGRID, etc.). The significance of the hypergeometric analysis is indicated as −Log$_{10}$ ($p$ value) in the form of a horizontal histogram where bar heights represent level of significance. Bars are color coded based on their inclusion in each cluster. Gene pathways or GO terms in different clusters, including polycomb repression complex 2 (PRC2) subunit, EZH2 (Ester of Zinc Finger Homolog 2), and SUZ12 (Polycomb Repressive Complex 2 Subunit)-binding sites significantly enriched in C3, suggesting hypermethylation of PRC2 de-represses gene expression. Genes in C1 have no significant gene set enrichment. **b, c** In silico analysis of EZH2 or SUZ12-binding sites within gene promoters in each cluster (C1–C4) and show more significant binding scores in C3 than in other groups, Box and whisker plot: center line, median (red) or mean (green); box limits, upper and lower quartiles; and whiskers, maximum and minimum values; red dots, outliers. Statistic significance was assessed by one-way ANOVA. **d** The probability of the top 20 transcription factor consensus-binding sites in C3 showed EZH2 has the highest binding scores in a subset of genes in C3, including WNT2. The heatmaps for C1, C2, and C4 are in Supplementary Fig. 18. **e** In non-canonical gene cluster C3 ($n = 67$), WNT2 is significantly hypermethylated in the promoter (one-way ANOVA, FDR = 6.6005e−03) and highly expressed in ESCC (one-way ANOVA, FDR = 0.0039). WNT2 also shows the highest fold change in gene expression in ESCC relative to adjacent normal tissues. Each dots represent individual gene; colored dots reflect different categories genes.

state. Thus, we also compared RNA-seq data from a normal cell line, Het-1A with that of the ESCCAL-1 knockdown tumor cells, EC9706. Using an iterative clustering approach, we identified 210 significant genes whose expression was differential between shControl EC9706 and shESCCAL-1 EC9706 cohort and was similar between shESCCAL-1 EC9706 and Het-1A (normal) cohort (gene list in Supplementary Data File 5)[45]. Functional annotation on these identified DEGs exhibit an enrichment of RNA binding, ribosomal proteins, and Myc target-gene sets (Fig. 7i, Supplementary Fig. 31). Myc knockdown depleted ribosomal translational processes in ESCC, phenocopying the effect of ESCCAL-1 knockdown (Fig. 7j). These results indicate that ESCCAL-1 participates in the biological process of ribosome regulation and Myc-mediated regulation of genes, which extends current knowledge on the potential role of translational machinery and Myc signaling in ESCC[46]. Moreover, simultaneously abolishing ESCCAL-1 and WNT2 expression increased apoptosis compared to individual knockdown in vitro (Supplementary Fig. 32). Thus, beyond WNT2-mediated WNT pathway activation, other aberrant signaling pathways activated by ESCCAL-1 upregulation also contribute to ESCC tumorigenesis. Therefore, epigenetic dysregulation promotes ESCC through divergent and multi-factorial mechanisms.

## Discussion

The development of ESCC is a complex, dynamic biological process that involves multiple steps of genetic and epigenetic alterations. Numerous genetic studies of ESCC revealed genomic alterations in genes within the cell cycle, p53, AKT/mTOR, and Hippo signaling pathways[4,6,8,47,48]. It remained unclear in ESCC, as in most other cancer types, whether and how the epigenetic landscape contributes to cancer pathogenesis. We performed WGBS, RNA-seq, and proteomic analyses on matched normal and tumor samples along with analysis of TCGA-ESCA datasets and validated our findings in independent samples. We observed 98% of global CpG hypomethylation and 2% of local CpG hypermethylation across the ESCC genome. The trend of global loss of DNA methylation is consistent with studies in colon and other types of cancers, but the degree of hypomethylation in ESCC appears to be significantly larger than reported in other cancers[12,49]. The characteristics of CpG methylation alterations can discriminate cellular states between tumor and normal conditions, and histological subtypes of esophageal cancer. DNA methylation is a defining feature of cellular identity and is essential for cell development[50]. Cancer-specific DMRs have been identified in colon cancer and such stochastic methylation variations distinguish cancer from normal and may serve as diagnostic or therapeutic biomarkers[13]. We found that the heterogeneity of DNA methylation alteration is greater in ESCC

relative to normal esophageal tissues. Higher variance of DNA methylation alteration in squamous carcinomas (ESCC, HNSC) is strongly associated with poor clinical outcome. Our findings provide insight into the potential clinical relevance of epigenetic dysregulation and heterogeneity as a molecular biomarker of clinical outcome in cancer.

We validated prominent epigenetically altered coding and non-coding genes from the non-canonical cluster (C3) and canonical cluster (C2), respectively. The WNT pathway is epigenetically dysregulated in ESCC by the inactivation of negative regulators (SFRP1/2/4/5, SOX17, and WIF1)[33]. Our multi-omics data identified high WNT2 expression along with a highly methylated promoter region in ESCC. We verified this observation in independent samples and ESCC cell lines. We experimentally demonstrated decreased EZH2 binding to the hypermethylated promoter region of WNT2 in cancer cells relative to normal cells, in association with higher expression of WNT2 in cancer. DNA methyltransferase inhibitor treatment in cancer cells reversed this phenotype. Our findings indicate that hypermethylation-mediated de-repression of WNT2 activates the WNT pathway in ESCC. TCGA-ESCC cohort suggested higher expression of WNT2 is a biomarker of tumor progression. Knockdown of WNT2 expression suppressed cancer cell growth, reduced cellular invasion and migration, reduced β-catenin target-gene expression (MMP3/9) in vitro, and inhibited xenograft tumor growth in vivo. Our data provide new insight into the mechanism of epigenetic dysregulation that results in non-canonical gene-expression regulation in cancer and the underlying molecular events promoting WNT pathway activation in ESCC.

LncRNA dysregulation is an emerging but poorly understood feature of oncogenesis[21]. We reported ESCCAL-1 overexpression in ESCC[21], which is also overexpressed in other cancer types[40–42]. Overexpression of ESCCAL-1/CASC9 promotes cancer cell growth[51], invasion[52], and metastasis[53]. We discovered that loss of methylation in its promoter and an increase of YY1 TF binding is a principle molecular mechanism of ESCCAL-1 dysregulation in ESCC, resulting in cell cycle and ribosomal pathway dysfunction. We found that ESCCAL-1 plays an epigenetic-mediated causal role in tumor growth and is a biomarker of worse clinical outcome in ESCC. ESCCAL-1 is also overexpressed in other cancer types and associated with drug resistance in lung cancer[40]. Whether ESCCAL-1 is similarly dysregulated by epigenetic mechanisms in other cancer types beyond ESCC remains to be investigated. Nevertheless, suppressing ESCCAL-1 expression, potentially using antisense RNA[54] or CRISPR-based strategies[55], may be a promising therapeutic approach in ESCC and other cancers.

Overall, our study provides a rationale and a roadmap for delineating the landscape and functional roles of epigenetic

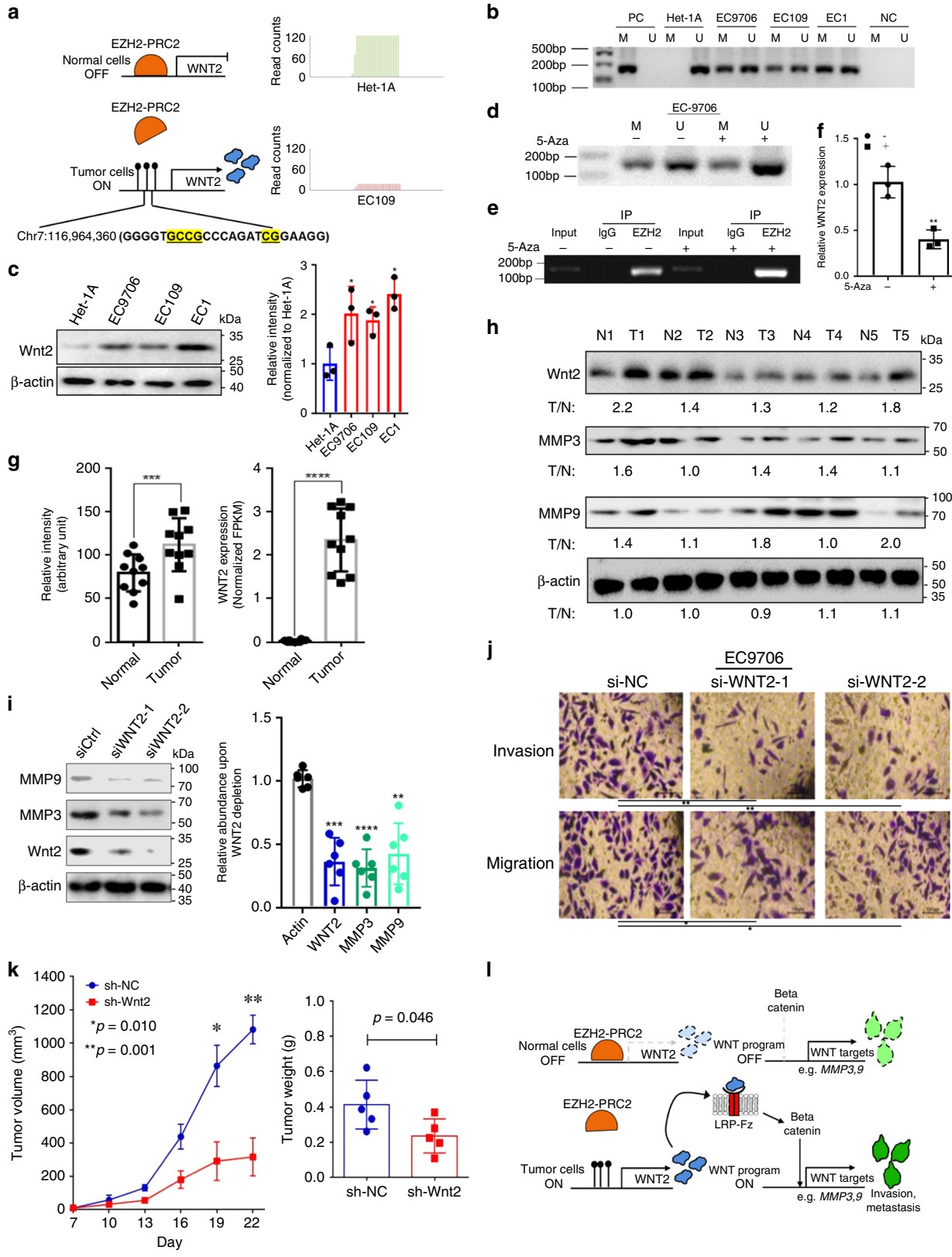

dysregulation in cancer at a genome-wide, high resolution. Further analysis will provide a better understanding of the impact of epigenetic dysregulation and heterogeneity on various cancer-associated phenotypes and treatment responses[14]. Multi-regional WGBS or single-cell DNA bisulfite sequencing could facilitate this opportunity in the future. Our study provides a resource that includes comprehensive profiling of the epigenetic landscape to enable the discovery of additional biomarkers and therapeutic targets in ESCC and potentially other cancers.

**Fig. 5 Hypermethylation in the WNT2 promoter leads to high WNT2 expression in ESCC. a** Chromatin immunoprecipitation by high-throughput DNA sequencing (ChIP-seq) showed EZH2 preferentially binding to the hypomethylated WNT2 promoter region in normal cells (Het-1A) relative to the hypermethylated WNT2 promoter region in ESCC cells (EC109). **b** The WNT2 promoter region is hypermethylated in ESCC cell lines by methylation-specific PCR (MS-PCR) analysis. M methylation detection, U unmethylation detection, PC positive control, NC negative control. Het-1A cells are an immortalized normal esophageal epithelial cell line. EC9706, EC109, and EC1 are patient-derived ESCC cell lines. **c** Western blot analysis showed WNT2 protein is overexpressed in three ESCC cancer cell lines compared to normal cell line Het-1A. **d** EC9706 cancer cells were treated with or without the DNA methyltransferase inhibitor, 5-Azacytidine (5-AzaC), for 24 h. The methylation change status in promoter region of WNT2 was detected by MS-PCR. **e** ChIP-PCR detection of EZH2 pull-down in EC9706 cancer cells were treated with or without 5-AzaC for 24 h. **f** RT-PCR detection of WNT2 expression in EC9706 cancer cells were treated with or without 5-AzaC for 24 h. **g** WNT2 is overexpressed in ESCC tumor ($n = 10$) and adjacent normal samples ($n = 10$). Left panel: protein expression from western blots, right panel: RNA abundance (FPKM) from RNAseq. **h** WNT2, MMP3, and MMP9 expression in the tumor and normal samples. Representative blots from 10 paired normal and ESCC tumor samples. **i** WNT2 depletion with two independent siRNAs inhibited MMP3 and MMP9 expression in the ESCC cell line EC9706. **j** Two independent siRNAs to silence WNT2 expression reduced the invasion and migration of ESCC cells relative to siRNA controls. Scale bar:100 μm. **k** Xenograft tumor growth curve and tumor weight in EC9706 cells expressing either shRNA control (sh-NC, $n = 5$) or sh-Wnt2 ($n = 5$). **l** Schematic representation of the mechanism of EZH2/PRC2-WNT2-MMP signaling upregulation in ESCC. Functional study of WNT2 in EC109 cell line is presented in Supplementary Fig. 27. Representative results from three independent experiments are presented in **b–e**, **j;** triplicates in each condition are shown in **f**. The bar plots are plotted as mean ± s.d. Statistic significance was measured using two-sided t-test. *$p < 0.05$, **$p < 0.01$. ***$p < 0.001$, ****$p < 0.0001$. Source data are provided as a Source Data file.

## Methods

**Primary ESCC specimens**. Matched clinical samples of esophageal squamous carcinoma and adjacent normal esophageal tissue are obtained from ten patients (Linzhou Cancer Hospital) as fresh frozen specimens at Translational Medical Center, Zhengzhou Central Hospital, affiliated to Zhengzhou University, China (Supplementary Fig. 1). Written informed consent was obtained from patients before surgery. The collection of human samples and the protocols for the investigations were under the approval of the Institutional Ethics Committee of Zhengzhou, Henan Province. These specimens were used for sequencing and different assays; WGBS ($n = 20$), WGS ($n = 6$), RNA-seq ($n = 20$), and iTRAQ proteomic assay ($n = 20$). For validation, independent 93 matched ESCC and adjacent normal samples with clinical follow-up ($n = 73$) were prepared for gene expression analysis.

**Whole-genome sequencing**. DNA was extracted using the QIAamp DNA Mini Kit (Qiagen), fragmented using Bioruptor® Pico. Libraries were constructed using VAHTSTM Universal DNA Library Prep Kit for Illumina V3 ND607-02 (Vazyme, Nanjing). Libraries were sequenced with an Illumina HiSeq 4000 to obtain 150 bp paired-end reads. Base calling was performed with the Illumina Real Time Analysis version 2.7.7 and the output was demultiplexed and converted to FastQ format with the Illumina Bcl2fastq v2.19.0.316. The FastQC package (http://www.bioinformatics.babraham.ac.uk/projects/fastqc) was used to check the quality of the sequencing reads. Sequencing adapters were trimmed from raw reads with Trimmomatic (version 0.36)[56]. We performed mapping, duplicates marking, and mutation calling with bcbio-nextgen (https://github.com/bcbio/bcbio-nextgen), a community developed platform for variant calling. Specifically, Reads were mapped to the human reference genome assembly GRCh37 using BWA-MEM in the BWA package (version 0.7.17) with the default parameters[57]. Duplicates were marked using the tool Sambamba version 0.6.6 (ref. [58]). Somatic mutations include single-nucleotide variants (SNVs), small insertions and/or deletions (indels), and structural variants (SVs). The detection of somatic mutations was performed using tumor and matched normal whole-genome BAM files generated in the steps described above. We used a series of software packages, including VarDict[59], MuTect2 (ref. [60]), and Strelka2 (ref. [61]), to detect somatic SNVs and indels, and packages including LUMPY[62], Manta[63], CNVkit[64], and MetaSV[65] to detect SVs.

**Whole-genome bisulfite sequencing**. Genomic DNA was extracted with QIAmp DNA Mini kits (Qiagen) from fresh frozen tissue samples. One microgram DNA was fragmented by sonication with a base pair peak of 300 bp for the resulting fragment, and adaptors were then ligated to both ends of the fragments. Bisulfite conversion was performed on whole genomes of ten pairs of ESCC and matched normal tissues, where converts cytosine residues of the dinucleotide CpG to uracil but leaves methylated cytosine unaffected[66]. PCR amplification and purification were carried out. The uracil-binding pocket of KAPA HiFi DNA polymerase has been inactivated, enabling amplification of uracil-containing DNA. The high quality of the library was estimated by The Qubit® 3.0 Fluorometer. Bisulfite conversion success ratio is 99.18% and 99.49%, respectively, in normal and ESCC samples (Supplementary Data File 1). The WGBS library was sequenced on an Illumina HiSeq2500 sequencers and generated 400M of paired-end reads (2× 125 bp).

**Whole-transcriptome sequencing (RNA-seq)**. The RNA was extracted with TRAzol from ten pairs of fresh frozen tissue samples. The input material for total RNA-seq library preparation was 2 μg per sample. Sequencing libraries were generated using NEBNext® Ultra™ RNA Library Prep Kit for Illumina® (#E7530L,

NEB, USA) following the manufacturer's recommendations. Index codes were added to attribute sequences to each sample. Briefly, mRNA was purified from total RNA using poly-T oligo-attached magnetic beads. Fragmentation was carried out using divalent cations under elevated temperature in NEBNext RNA First Strand Synthesis Reaction Buffer (5×). First-strand cDNA was synthesized using random hexamer primer and RNase H. Second-strand cDNA synthesis was subsequently performed using buffer, dNTPs, DNA polymerase I, and RNase H. The library fragments were purified with QiaQuick PCR kits and elution with EB buffer. Terminal repair, A-tailing, and adapter ligation were implemented. The aimed products were retrieved by agarose gel electrophoresis followed by PCR, and then the library was completed. RNA concentration of library was measured using Qubit® RNA Assay Kit in Qubit® 3.0 to preliminary quantify and then diluted to 1 ng/μl. Insert size was assessed using the Agilent Bioanalyzer 2100 system (Agilent Technologies, CA, USA), and qualified insert size was accurately quantified using StepOnePlus™ Real-Time PCR System (Library valid concentration >10 nM). The clustering of the index-coded samples was performed on a cBot cluster generation system using HiSeq PE Cluster Kit v4-cBot-HS (Illumina) according to the manufacturer's instructions. After cluster generation, the libraries were sequenced on an Illumina Hiseq 4000 platform to 2 × 150 bp paired-end reads.

**Proteomic assay and data analysis**. Reduced and tryptic digested peptides from samples were labeled with eight isobaric ItraQ reagent for an individual run and mixed at an equimolar ratio. Resuspended labeled peptides were pH optimized and separated through strong cation ion chromatography. Samples prepared as such were run through reverse phase liquid chromatography–mass spectrometry (LC-MS). The isobaric labeling and LC-MS quantifications were operated in Beijing Genomics Institute (BGI) using optimized quantitative MS-MS protocols[67]. For Protein identification and data analysis, IQuant was used[68]. For improved protein identification, a Mascot Percolator and Mascot Parser, a customized post-processing tool was used. The signal-to-noise ratio was decreased by variance stabilization normalization. Due to the low abundance or low ionization of peptides, missing the reporter ions is a common phenomenon in isobaric data, and may hinder downstream analysis. A missing reporter was imputed as the lowest observed values to avoid estimation bias. Nonunique peptides and outlier peptide ratios are removed before quantitative calculation[73]. The weight approach proposed is employed to evaluate the ratios of protein quantity based on reporter ion intensities[69]. The ratio between normal and tumor samples were generated for each match control pairs. This way three distinct datasets were generated for ten pairs of tumor and normal samples. Sample number 7 (for both tumor and normal, T7 versus N7) was run in each time to standardize among three datasets. For dataset integration, each dataset was normalized by the T7/N7 ratio for the abundance of the protein in all datasets. Dataset normalized as such was represented as a matrix so that protein abundance (row-wise) can be compared across ten different samples as tumor versus normal quantitative ratios (column-wise) generated from three separate runs. The raw data in Supplementary Dataset file 6.

**Chromatin immunoprecipitation with DNA sequencing (ChIP-seq)**. ChIP assays were performed according to the protocol supplied with the kit (catalog no. 9003) from Cell Signaling Technology. Briefly, EC109 and Het-1A cells were cross-linked with 37% formaldehyde at a final concentration of 1% at room temperature for 10 min. Fragmented chromatin was treated with nuclease and subjected to sonication. Chromatin immunoprecipitation was performed with anti-KMT6/EZH2 antibody (ab195409, Abcam), Anti-Histone H3 (acetyl K27) antibody ChIP Grade (ab4729, Abcam), Anti-YY1 antibody (ab38422, Abcam), rabbit anti-histone H3 (a technical positive control; 1:50) (catalog no. 4620; Cell Signaling

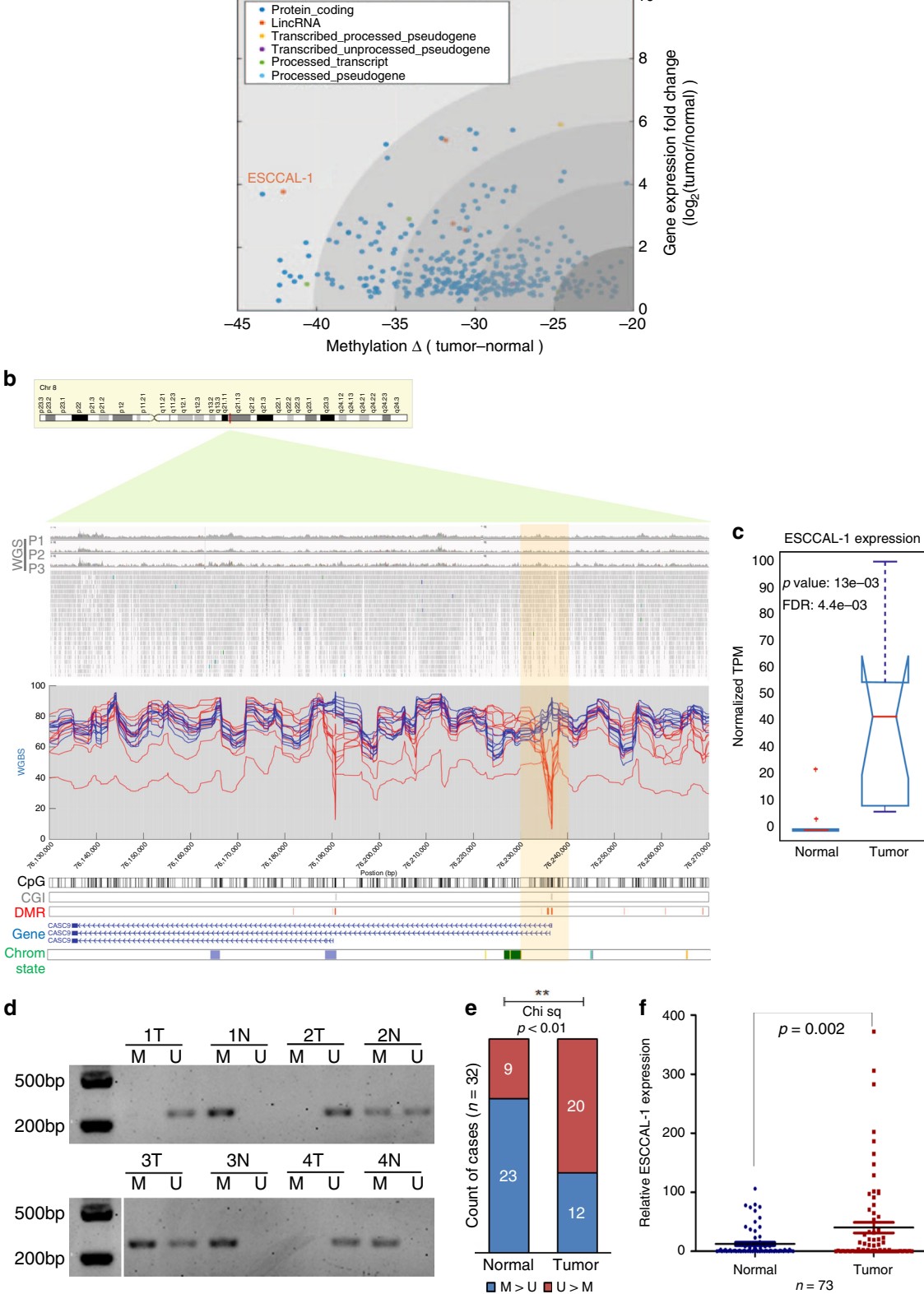

Technologies), and normal rabbit IgG (a negative control; 5 μg) (catalog no. 2729, Cell Signaling Technologies). After reverse cross-linking and DNA purification, ChIP-Seq libraries were prepared and sequenced on a HiSeq 4000 sequencer (Illumina, San Diego, CA). To ensure the accuracy of subsequent bioinformatics analysis, the original sequencing data were filtered to obtain high-quality sequencing data (clean data). Quality control of the sequencing data was performed using Sickle (https://github.com/najoshi/sickle) and SeqPrep (https://github.com/jstjohn/

SeqPrep). The sequencing output raw reads were trimmed by stripping the adaptor sequences and ambiguous nucleotides and reads with quality scores less than 20 and lengths below 20 bp were removed. The cleaned reads were aligned to human reference genome hg19 using BWA. MACS2 (model-based analysis of ChIP-seq) algorithm[70] was used for peak calling. The reads of EZH2 binding on WNT2 promoter region were visualized using Integrative Genomics Viewer (IGV, Broad Institute).

**Fig. 6 Hypomethylation-mediated upregulation of long non-coding RNAs (lncRNAs) in ESCC. a** In canonical gene cluster C2 ($n = 389$), ESCCAL-1 is significantly hypomethylated in the promoter (one-way ANOVA, FDR = 1.7386e−04) and highly expressed in ESCC (one-way ANOVA, FDR = 0.01). ESCCAL-1 also shows the most significant and substantial methylation difference between normal esophageal tissues and ESCC among the lncRNAs in C2. Each dots represent individual gene; colored dots reflect different categories genes. **b** Loss of CpG methylation at the ESCCAL-1 promoter region in ESCC (chr8:76,135,639–76,236,976 of GRCh37/hg19). WGS (whole-genome sequencing) of three ESCC patients shows no mutation or copy number variations detected at the above indicated region. This is validated by TCGA ESCA data ($n = 186$), where no no mutation or copy number variations were observed (see Supplementary Fig. 24d). DMRs around transcription start sites (TSSs) of the two isoforms showed extensive differentiation between ESCC and normal samples. **c** ESCCAL-1 was significantly differentially expressed and highly abundant in ESCC samples ($n = 10$) relative to normal samples ($n = 10$). Statistic significance was assessed by one-way ANOVA, $p$ value = 0.0013, $\log_2$ (fold change) >1, FDR = 0.0044. **d** Methylation status of the ESCCAL-1 promoter region was verified on an independent matched normal ($n = 32$) and ESCC tumor ($n = 32$) samples using a methylation-specific PCR (MS-PCR) assay; four representative PCR results are shown. M: PCR with methylation primers, U: PCR with unmethylation primers. **e** Quantification of MS-PCR results in the 32 paired normal and tumor samples. Chi square analysis tested for significance between groups. P value < 0.01. **f** ESCCAL-1 expression is significantly higher in ESCC tumors in an independent cohort of 73 matched normal and tumor samples. Error bars are median values with 95% confidence intervals. Significance for comparison between the two cohorts was measured using an unpaired two-sided Student's $t$-test, $p = 0.002$.

**WGBS data preprocessing**. Bisulfite-treated DNAs are further sequenced using Illumina HiSeq 2500 system. Approximately 400 M of paired-end reads (125 bp x 2) and 100 Gbp per sample are generated except one sample (N15). The reads are sent out to the pipeline. Our pipeline consists of four steps (Supplementary Fig. 2a). Firstly, we trimmed using Trim Galore! (v0.4.1) to remove Illumina adaptors with the options of "--paired –length 50 --clip_R1 6 --clip_R2 6". After trimming, around 90 Gbp of data remained per sample. Then trimmed reads were aligned to the HG19 reference genome using BSMAP (v2.89) with the option of "-p 8 -R". Then SAMtools (v. 1.3.1) was used to sort by genomic coordination and make a bam file index. Picard Tools (v.1.92) is used to remove PCR duplicates. After deduplication, ~70 Gbp remained per sample. Lastly, we ran MOABS (v. 1.3.4) to compute the methylation ratio per CpG with the option of "--cytosine-MinScore 20 --skipRandomChrom 1 -p 4 --keepTemp 0 --processPEOverlapSeq 1 --requiredFlag 2 --excludedFlag 256 --minFragSize 110 --reportCpX G --qualityScoreBase 0 --trimRRBSEndRepairSeq 0 --trimWGBSEndRepairPE1Seq 5 --trimWGBSEndRepairPE2Seq 5". Around 95% of CpGs are covered by at least five reads.

**WGBS data matrix**. Given the methylation ratio and coordination computed, we built a data matrix, whose columns are samples and rows are CpGs. The normal tissue WGBS of sample 15 (N15) has generated a small amount of volume (~4.5 Gbp), which is just 6% compared to average (72 Gbp) and covers the half of CpGs with notably low coverage (1.58×) than other samples (~15×). Thus, we decided to eliminate N15 for the further downstream analysis. As a result, we have ten ESCC samples and nine normal esophageal tissue samples. For more robust analysis, we applied the minimum threshold 5× and also selected CpGs that all samples have its methylation ratio. This screening process gave 13 M of CpGs with confident methylation ratio (Supplementary Data File 1).

**Cross-validation between WGBS and TCGA ESCA**. We aligned our data with TCGA ESCA data ($n = 202$). The methylation data of TCGA ESCA exploited Illumina Infinium Human Methylation 450K Beadchip to measure methylation level for 202 samples: 186 esophageal cancer tissue samples and 16 esophageal normal tissue samples. To compare TCGA ESCA HM 450K methylation ratio and WGBS methylation ratio of our data, we computed the mean methylation ratio of the tumor and normal samples per CpG for both TCGA and our WGBS data. Around 300K of CpGs are in the intersection of TCGA ESCA HM 450K and WGBS. We computed the Pearson correlation coefficient (PCC) to measure the representative power in our dataset albeit a rather small sample size. First, we calculated the PCC of the tumor and normal from TCGA ESCA HM 450K and WGBS. The highest PCC (=0.9674) is between TCGA ESCA normal and WGBS normal since normal tissues are relatively homologous. The PCC (0.9639) between TCGA ESCA tumor and WGBS tumor was followed due to tumor heterogeneity but still showed a high correlation. The third and fourth PCCs are between TCGA ESCA tumor and WGBS normal (=0.9512), and TCGA normal and WGBS tumor (=0.9468) due to the difference between tumor and normal esophageal tissues.

We downloaded clinical annotation to match the histological type of each sample. All of the WGBS data is ESCC. The PCC between TCGA ESCC and WGBS ESCC is 0.757, the PCC between TCGA EAC and WGBS ESCC is 0.5554 (Supplementary Fig. 3).

**Data processing of RNA-seq**. RNA-Seq reads were mapped to the HG19 reference genome using STAR (Spliced Transcripts Align to a Reference, v2.4.2a). The expression level of transcript per million (TPM) reads were quantified using RNA-Seq by Expectation-Maximization algorithm (RSEM v1.2.29). The quantified gene expressions of 26,334 transcripts (including coding genes and non-coding genes) were processed in Rstudio console with R programme (v 3.4). Differentially expressed genes between tumor and normal samples were identified using the EdgeR algorithm.

**Differentially methylated CpG (DMCs)**. Among 18,421,444 of CpGs, we computed the $F$-statistics from one-way analysis of variance (ANOVA) to identify confident CpGs. Almost half of them has very low $p$ values (<0.05). The $p$ values are further adjusted by Benjamini–Hochberg procedure to compute FDR. In total, 5,092,845 of CpGs has $q$ values less than 0.05. The 5,092,845 of DMCs is used for the downstream study in this paper. Among 5,092,845 of DMCs 97.29% are hypomethylated in ESCC samples while only 2.71% are hypermethylated in ESCC samples.

**Entropy analysis**. Entropy is computed per CpG in both ESCC and normal esophageal cohorts separately as a measure of variance. The "entropy" function was used in the "stats" package of SciPy (v 0.19.1) on top of python3 (v3.5.2). The bin size was 10%. The distribution of CpG entropy was plotted using MatLab (v. 9.2) "plot" and "histogram" function with the default option.

**DMC enrichment analysis with genomic annotation**. We annotated the regulatory elements of the confident DMCs. The genomic coordination of exons and introns were downloaded from UCSC Genome Browser (http://hgdownload.soe.ucsc.edu/goldenPath/hg19/database/refGene.txt.gz). The promoter is computed 4.5 kbp upstream and 500 bp downstream given TSS. The genomic coordinates of enhancers are downloaded from VISTA Enhancer Browser (https://enhancer.lbl.gov). We learned that hypomethylated DMCs outnumber hypermethylated DMCs in ESCC, 4.95M versus 138K, respectively. Among hypermethylated in DMCs in ESCC, 83.67% has overlapped with regulatory elements while only 56.77% of hypomethylated DMCs has overlaps. Such overlaps were further dissected into enhancers, promoters, exons, introns, and CpG islands (Supplementary Fig. 4).

**DMC enrichment analysis with functional annotation**. We performed functional annotation on the confident DMCs except for protein-coding RNAs. Functional annotation includes long noncoding RNA (lncRNA), antisense RNA, MicroRNA (miRNA), small nuclear RNA (snRNA), small nucleolar RNA (snoRNA), small cytoplasmic RNA (scRNA), ribosomalRNA (rRNA), vaultRNA, and Mt_tRNA. The functional annotation was downloaded from the GENCODE project (https://www.gencodegenes.org/releases/27lift37.html). The composition of the functional annotation is illustrated (Supplementary Fig. 5). We deconvoluted functional mapping of hypermethylated DMCs and hypomethylated DMCs. The majority of the hypermethylated DMCs are mapped to antisense RNAs (58.01%) followed by lncRNA (39.28%) while that of the hypomethylated DMCs are mapped to lncRNAs (63.08%) followed by antisense RNA (29.89%). The permutation test was used to perform statistical analysis with regioneR package (Supplementary Fig. 5).

**Transcription factor-binding site analysis**. Transcription factors and their binding sites annotation were downloaded from the ENCODE project (http://hgdownload.cse.ucsc.edu/goldenpath/hg19/encodeDCC/wgEncodeRegTfbsClustered/). First of all, we computed the proportion of each TF. With regard to base-pair counting, POLR2A, the largest contributor to the composition, followed by CTCF (Supplementary Fig. 11a). We mapped DMCs to TF-binding sites and calculated the composition of TFs both in DMCs and HG19. Enrichment was computed as a ratio of the proportion of TFBS mapped to DMCs over TFs in HG19 (Supplementary Fig. 11b). Relatively DMC are most enriched in SUZ12 followed by EZH2, which are the component of the Polycomb Repressive Complex 2 (PRC2) (Supplementary Fig. 11c). Methylation type has been studies among top 20 TFs that DMCs are enriched. Hypomethylated DMCs dominated in the most TFs except for three TFs: SUZ12, EZH2, and CTBP2, which is related to endometrial cancer pathway and WNT pathway (Supplementary Fig. 11d).

**TSS methylation level analysis**. The methylation ratio was summarized in every 200-bp window relative to TSS. Then the methylation ratio per bin was normalized

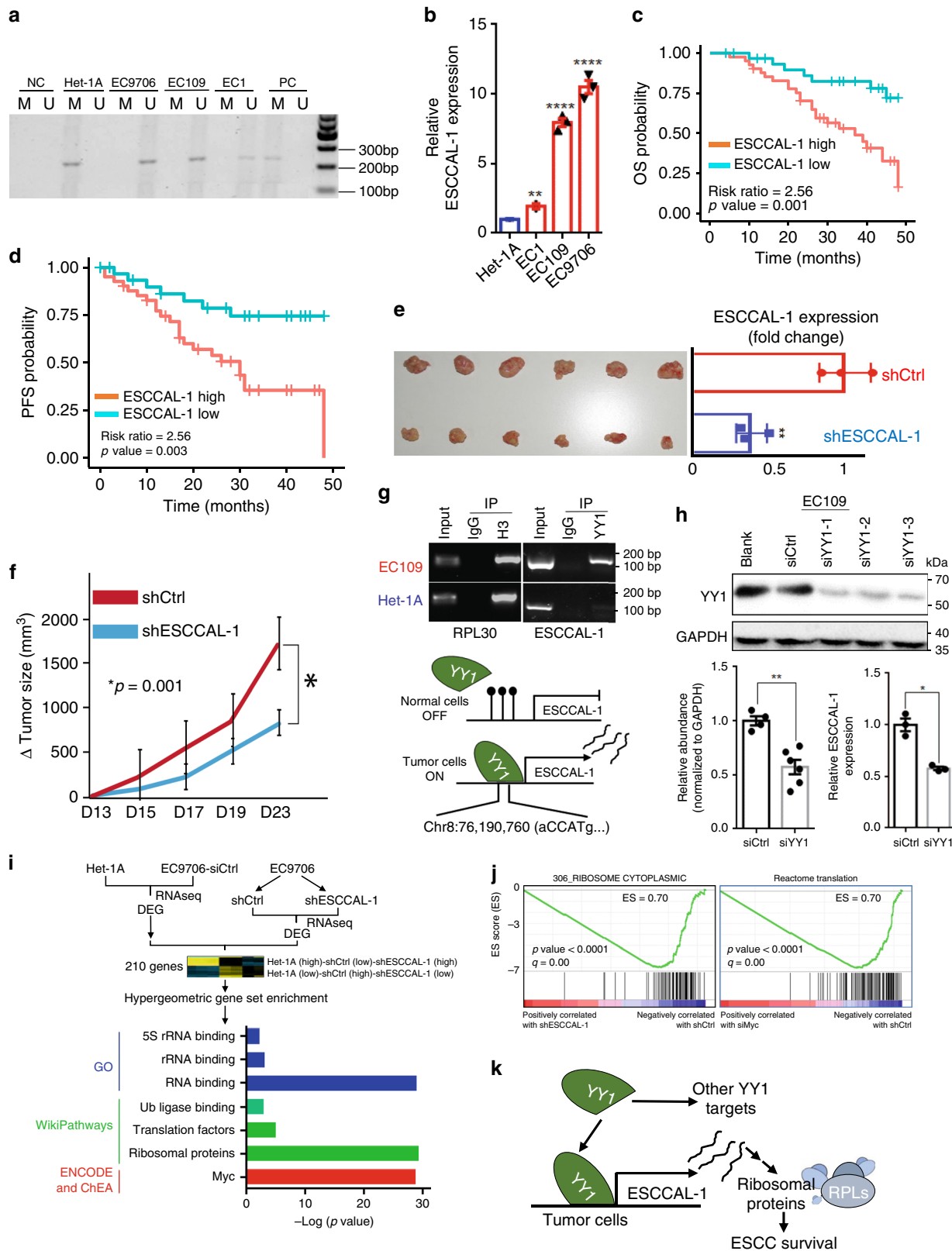

and averaged for both ESCC and normal cohorts. Methylation conversion is observed both upstream and downstream given TSS. The normalized methylation ratio of ESCC samples is higher between around −3000 to 7000 bp given TSS. The graph was made by R with spline interpolation with default options.

**Differentially methylated regions.** DMRs were computed from 5,092,845 of confident DMCs. The window size is flexible as long as any two CpGs locate in 150 bp and have consistent methylation pattern either keeping hypermethylated or

hypomethylated. The criteria make sure the minimum CpG density is at least 0.01. DMRs peak size is of 150–350 bp and CpG density peak is of 0.04–0.05 (Supplementary Fig. 8a, b). The genomic region enrichment analysis for DMRs was conducted using LOLA package (http://code.databio.org/LOLA) in R.

**Integrating and clustering methylome and transcriptome data.** To integrate methelomic and transcriptomic data, we focused on methylation in the promoter regions which are defined 4.5 kbp upstream and 500 bp downstream given TSS. We

**Fig. 7 Oncogenic functions of ESCCAL-1 in ESCC. a** Hypomethylation at ESCCAL-1 promoter regions was confirmed in three different ESCC cell lines using methylation-specific PCR. M: PCR with methylation primers, U: PCR with unmethylation primers. Het-1A: an immortalized esophageal epithelial cell line. EC1, EC109, and EC9706 are patient-derived ESCC cell lines. **b** ESCCAL-1 expression was significantly higher in three ESCC cell lines (EC1, EC109, and EC9706) relative to a normal esophageal epithelial cell line (Het-1A). Triplicates in each cell lines, mean ± s.d., unpaired two-sided t-test, **$p < 0.01$, ****$p < 0.0001$. **c, d** ESCC patients with higher expression of ESCCAL-1 exhibit worse overall (OS) and progression-free survival time (PFS), risk ratio = 2.56, log-rank test, p value = 0.003. **e, f** shRNA knockdown of ESCCAL-1 inhibited tumor growth in a tumor xenograft mouse model ($n = 6$ in each group, unpaired two-sided t-test, p value = 0.001). ESCCAL-1 remains greater than 50% lower expression in xenograft tumors of ESCCAL-1 knockdown group relative to the control group. Bar plots (mean ± s.d.) indicate triplicates in each condition for RT-PCR. **$p < 0.01$ was measured by unpaired two-sided t-test. **g** Chromatin immunoprecipitation by YY1 transcription factor protein-directed antibody followed by a standard polymerase chain reaction (PCR) assay in ESCC cancer cells (EC109) and normal esophageal cells (Het-1A). IgG was used as negative control. H3 was used as a positive control. Representative results from three independent experiments. **h** Representative western blot assay showed decreased YY1 protein expression following 72 h post-transfection of three independent siRNAs targeting various transcript regions of YY1 in ESCC cell line (EC109). ESCCAL-1 expression was measured in YY1 knockdown cells using RT-PCR. Bar plots (mean ± s.d.) showed quantification of YY1 protein expression from two independent experiments or ESCCAL-1 expression from triplicates in each condition. Statistical significance was assessed by unpaired two-sided t-test, *$p < 0.05$, **$p < 0.01$. **i** RNAseq was performed in duplicate in the normal esophageal cell line Het-1A, ESCC cancer cells EC9706 with control shRNA, or EC9706 with an shRNA against ESCCAL-1. Unsupervised hierarchical clustering of differential gene clusters between the three conditions is shown. Differentially expressed genes were selected based on an iterative clustering approach selecting for genes with the top 5% of the most variable and differential gene expression. Two hundred and ten genes were identified and subjected to functional annotation using a hypergeometric test in multiple databases. **j** Gene Set Enrichment Analysis (GSEA) for RNA-seq data from either ESCCAL-1 knockdown or Myc knockdown in EC9706 showed ribosomal genes (pathways) enriched in EC9706 cells. **k** Diagram illustrating YY-1 binding to hypomethylated promoter regions of ESCCAL-1, driving its overexpression and leading to dysregulation of ribosomal genes and ESCC progression. Source data are provided as a Source Data file.

aligned DMRs that are identified previously with q value <0.01 to the promoter regions of all genes. We also increased the promoter size up to 11 kbps to see if any level of extension of promoter size can affect the final gene set whose promoters hold any DMRs, but that added only for more genes. As a result, we learned that 4391 genes have significant DMRs in their promoter regions. High-throughput transcriptome sequencing (RNA-Seq) was conducted. Around 20,000 gene expression levels are estimated for both normal and ESCC pairs of ten patients. Significantly differentially expressed transcriptomes between two cohorts are selected given absolute log2(fold change) ≥1 and q value <0.05. In total, 4768 genes are significantly highly expressed in the ESCC cohort or the normal cohort. The number of genes in the intersection, i.e. the genes whose promoters are differentially methylated and expressions are significantly different between normal and ESCC cohorts, is 694 in total. We categorized those genes into four groups (1) C1: hypermethylated in their promoters with low gene expression in ESCC; (2) C2: hypermethylated in their promoters with high gene expression in ESCC; (3) C3: hypermethylated in their promoters with high gene expression in ESCC; (4) C4: hypomethylated in their promoters and low gene expression in ESCC. The genes in the former two groups (C1 and C2) are explained by canonical promoter methylation and gene expression model while the genes in the latter two groups (C3 and C4) are not. The gene list of the four groups is in Supplementary Data File 4. The heat map with dendrograms is made using "clustergram" command of MatLab v9.4 with parameters of "Standardize", "Row", "Colormap", "redgreencmap".

**CTCF analysis in gene promoter and gene body.** The genomic coordination of CTCF was retrieved from ENCODE Project (https://www.gencodegenes.org/releases/27lift37.html). We identified the number of CTCF-binding site overlapped with promoter regions by BEDTools (v2.26.0) with "intersect -wa -wb -a -b" option. The CTCF ratio in the gene promoter region is highest in C3.

**Gene body methylation level analysis.** DMCs (FDR ≤ 0.05) in the gene body and promoter region is selected for C1 and C3 to search hypothesis of noncanonical correlation in C3, where promoters are hypermethylated with high gene expression. Overall gene body is hypomethylated for both C1 and C3, but such reverse methylation between promoter and gene body is much prominent in C1 genes.

**Copy number alteration inferred from RNAseq.** CNVkit-RNA[71] was used to infer copy number alterations from RNAseq reads. The segments and recurrent copy number gains or loss across samples were generated and plotted using GISTIC 2.0 algorithm (Supplementary Fig. 15a, b).

**Simulation of CpG methylation heterogeneity.** Simulations of Ornstein–Uhlenbeck processes have been performed for CpG islands at selected promoter regions in both normal and cancer samples[17]. The model is described as $\theta(\mu - M)dt + \sigma dW$, with $M$ being the methylation value, $\mu$ the equilibrium point, $\theta$ the restoring force, $\sigma$ the noise level (4%), and $dW$ a Wiener process increment. Shown are ten example traces of simulated methylation levels. Regulatory forces ($\theta$) are set high in the normal tissue and low after a carcinogenic event. For the data in this study, the model works well for regions with single-peaked or broad methylation distributions. The histogram shows methylation variation by the stochastic simulation if more samples collected (Supplementary Fig. 6e).

**Co-expression analysis of RNAseq.** Co-expression analysis was conducted in R environment using RedeR package[72]. Co-expression analysis is to compute a null distribution via permutation and returning the significant correlation values. In all, 1000 permutations were performed to build the null distribution with Pearson correlation. We considered correlations with p value less than 0.01 with FDR adjustment as significant. The hierarchical clustering analysis used the complete method and considers the distances of each individual component to progressively computing the clusters until it finds a stable state. The final result is a dendrogram presenting hierarchical leaves, which has been used to plot the network. To clear the visualization, clusters ware nested using the fourth level of dendrogram to build the nests.

**Methylation-specific PCR (MS-PCR).** DNA was extracted from cells using the AllPrep DNA mini kit (Qiagen) according to the manufacturer's instructions and was quantified by NanoDrop analysis. Bisulfite modification was carried out on 200–500 ng of DNA using the EZ DNA Methylation-Gold Kit (Zymo Research) according to the manufacturer's instructions. MS-PCR analysis was carried out using a BioRad T100TM Thermal Cycler with 20 µl reaction mixtures. Primers for modified methylated sequences and modified unmethylated sequences are listed below. The PCR reactions were carried out under the following conditions: 95 °C for 10 min, 95 °C for 30 s, 50 °C for 30 s, and 72 °C for 30 s for a total of 45 cycles, 72 °C for 10 min. Five microliters of PCR product were used for electrophoresis on 2% agarose gel, and the methylated strip and unmethylated strip were analyzed by gel imaging analyzer. Representative results showed the ECSSAL-1 promoter methylation status identified by MS-PCR in ESCCs. Lanes M and U indicate the amplified products with primer recognizing methylated and unmethylated sequences, respectively. NC, negative control; PC, positive control. Primers for MS-PCR are listed in Supplementary Table 1.

**Functional annotation analysis.** Curated gene sets were collected from multiple curated databases such as ENCODE, CHEA, EnrichR, KEGG, WikiPathways, Reactome, GO molecular function, Panther, and BIOGRID. Overlapping between differentially methylated and/or expressed genes were estimated for significance using hypergeometric statistics. P values were adjusted to Bonferroni correction. $-\log_{10}(p$ value) for significance was measured and compared.

**Cell cultures.** Human ESCC cell lines (EC109, EC9706, EC1) and immortalized esophageal epithelial cell line Het-1A were purchased from the Shanghai Institutes for Biological Science (Shanghai, China). All cell lines were cultured in Dulbecco's modified Eagle's medium (DMEM) supplemented with 10% fetal bovine serum (Hyclone, Logan, UT, USA) and maintained at 37 °C in a humidified 5% $CO_2$ incubator.

**Cell transfection.** ESCCAL-1 shRNA, WNT2 siRNA, YY1 siRNA, Myc siRNA, and their control shRNA or siRNAs were synthesized from GenePharma (Shanghai, China). EC109 and EC9706 cells were transfected using Lipofectamine™ 2000 (Invitrogen, Carlsbad, CA, USA) according to manufacturer's instructions. Transfection efficacy was validated by RTq-PCR or western blot assay.

**Transwell migration and invasion assay**. Cell migration and invasion assays were carried out in transwell chambers (Costar, Lowell, MA, US) inserted into 24-well plates. For invasion assay, the upper chamber of the transwell plate was coated with matrigel and allowed to solidify at 37 °C and 5% $CO_2$ incubator for 30 min. The transfected EC109 and EC9706 cells were trypsinized, resuspended in serum-free culture medium, and adjusted to $2 \times 10^6$ cells/ml, added 200 µl cell suspension and 500 µl DMEM containing 10% fetal bovine serum to the lower chamber. After incubation for 48 h, the transwell chamber was fixed with 10% methanol, followed by the staining with crystal violet. Cells were counted under an inverted microscope. The protocol used for migration assay was similar to the invasion assay without the need to coat the upper chamber of the transwell with matrigel. Each experiment was conducted in triplicates.

**Western blot analysis**. Total proteins were extracted from tissues or cells using RIPA lysis buffer (Solarbio, China) containing protease inhibitor and then denatured at 100 °C for 10 min. Equal amounts of proteins in each group were separated by 10% SDS-PAGE. After PVDF membrane transfer, blocking in 5% skim milk and incubation with primary antibodies (MMP3, CST, USA; MMP9, CST, USA; Wnt2, Abcam, USA; β-actin, Santa Cruz, USA) and secondary antibody, respectively, the immunoreactive bands on the membrane were incubated with ECL kit and detected by using the Chemidoc EQ system (BioRad, USA).

**Mouse xenograft experiment**. Six-week-old male BALB/c immunodeficient mice were purchased from the Shanghai Experimental Animal Center, Chinese Academy of Sciences (Shanghai, China). Animal experimental procedures were carried out according to the Ethical Committee of Zhengzhou University. Mice were housed under a 12 h light/dark cycle and automatically given food and water. The EC9706 cells expressing with ESCCAL-1-shRNA, WNT2-shRNA, or shControl were subcutaneously injected into back flank of mice as the knockdown group ($n = 5$ for WNT2, $n = 6$ for ESCCAL-1) or the control group ($n = 5$ or 6). The tumor volumes were calculated as length × width$^2$ × 0.5 from day 13 to day 22 or 23 every two days, the mice were sacrificed at day 22 or 23 after injection.

**ChIP-PCR**. Chromatin immunoprecipitations were performed using digested chromatin from EC109 cells or Het-1A cells and the indicated antibody YY1. The antibody Histone H3 (D2B12) as a positive control. Purified DNA was analyzed by standard PCR methods using SimpleChIP® Human RPL30 Exon 3 Primers and ESSCAL-1 primers. Equal amounts of total genomic DNA (Input) were used for immunoprecipitation in each condition. Primer sequence information are given in Supplementary Table 1.

**Immunohistochemistry for WNT2 expression**. Immunostaining was performed on the paraffin-embedded tumor and adjacent normal tissues from ESCC patients' surgical removal samples. The avidin-biotin-peroxidase method was used to determine the location and relative expression level of the proteins. In briefly, xylene is used for paraffin section dewaxing, citric acid antigen repair buffer is used to repair the antigen, blocked endogenous peroxidase by 3% $H_2O_2$ and blocked with 3% BSA, and incubated overnight at 4 °C with primary antibodies WNT2 (1:100 dilution, Bioworld, USA). Then, secondary antibody was incubated at room temperature for 50 min, DAB stained and hematoxylin restained nucleus. Sections were visualized under a microscope at ×400 or ×200 (Olympus, Japan).

**Statistics**. Student's *t*-test was used for two group mean comparison. Wilcoxon rank-sum test was used for non-parametric mean comparison. One-way ANOVA was used for multiple groups comparison. Hypergeometric test was used in GSEA. Permutation test was used for comparison between observed target versus random evaluation. Multiple hypothesis test was adjusted with the Benjamini–Hochberg method. The standard $\alpha = 0.05$ was used as cutoff, and the null hypothesis is rejected when $p$ value <0.05. Different significant levels were used: *$p$ value < 0.05; **$p$ value < 0.01; ***$p$ value < 0.005; ****$p$ value <0.001. The 95% confidence interval (CI) for the median duration of progression-free survival and overall survival were computed with the robust nonparametric Brookmeyer and Crowley method. Hazard ratio with 95% CI and $p$ values were calculated with the Cox proportional-hazards regression model with survival package in R.

**Computational resources and code sharing for reproducibility**. Most of the analysis was done using SCG4 cluster of the Genome Sequencing Service Center by Stanford Center for Genomics and Personalized Medicine Sequencing Center. The SCG4 cluster has 20 compute nodes, each with 384 GB RAM, 56 CPUs each, and 40 compute nodes with 16 and 48 CPUs and 10 GbE connectivity. It shares 4+ PB of storage NIH dbGaP compliant and have 350+ software packages installed.

**Reporting summary**. Further information on research design is available in the Nature Research Reporting Summary linked to this article.

## Data availability

The WGBS, RNAseq, and ChIP-seq sequence data that support the findings of this study have been deposited in NCBI GEO database with the accession codes, GSE149608, GSE149609, GSE151838. WGS has been deposited in NCBI sequencing read archive (SRA) database with the accession code PRJNA630082. The proteomic data have been deposited in ProteomXchange database with the accession code, PXD019834. All the other data supporting the findings of this study are available within the article and its supplementary information files and from the corresponding author upon reasonable request. A reporting summary for this article is available as a Supplementary Information file. Source data are provided with this paper.

## Code availability

Code for WGBS analysis is available upon request.

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

## Acknowledgements

This study was supported by the National Natural Science Foundation of China (Grants 81171992, 31570899), the Natural Science Foundation of Henan (Grants 182102310328, 162300410279, 182300410374, 192102310096), the Education Department of Henan Province(18B310022,19A320037). This work used the Genome Sequencing Service Center by Stanford Center for Genomics and Personalized Medicine Sequencing Center, supported by the grant award NIH S10OD020141. E.C. acknowledges funding support from NCI Grants R01 [CA178015, CA222862, CA227807, CA239604, CA230263] and U24 [CA210974]. T.G.B. acknowledges funding support from NIH/NCI U01CA217882, NIH/NCI U54CA224081, NIH/NCI R01CA204302, NIH/NCI R01CA211052, NIH/NCI R01CA169338, and the Pew-Stewart Foundations.

## Author contributions

W.W. and W.C. conceived the project. W.W., W.C., H.L., A.Z., and S.M. performed WGBS, RNA-seq, and proteomic data analysis and construct the figures and manuscript. M.S., J. Chen, and N.S.A. helped WGBS and multi-omics analyses. G.S. performed simulation; A.H.S. conducted SNV analysis from RNA-seq; and Q.X., Y.C., J.L., H.G., P.L., X.S., L.S., P.H., and Y.L. performed experiments. J.W. and M.Y. collected, processed specimen as well as performed some experiments. H.X., W.T., J. Chang, J.G., and Y.G. performed whole-genome sequencing (WGS), whole-genome bisulfite sequencing (WGBS), whole transcriptome sequencing (RNA-seq), and isobaric tag for relative and absolute quantitation (iTRAQ). F.G.B. and E.C. helped to discuss the results. T.G.B. and M.S. guided, discussed, and edited the manuscript.

## Competing interests

E.C. is consultant at Takeda, Merck, Loxo, and Pear Diagnostics, reports receiving commercial research grants from AstraZeneca, Ferro Therapeutics, Senti Biosciences, Merck KgA and Bayerand stock ownership of Tatara Therapeutics, Clara Health, BloodQ Guardant Health, Illumina, Pacific Biosciences and Exact Biosciences. T.G.B. is an advisor to Revolution Medicine, Novartis, Astrazeneca, Takeda, Springworks, Jazz, and Array Biopharma, and receives research funding from Revolution Medicine and Novartis. M.S. is Cofounder and scientific advisory board member of Personalis, SensOmics, Mirvie, Qbio, January, Filtricine, and Genome Heart. He serves on the scientific advisory board of these companies and Genapsys and Jupiter. Other authors declare no competing interests.
