## [Peer Review File · Nature Communications]

Reviewers' comments:

Reviewer #1 (Remarks to the Author):

I have no further concerns. the authors have carefully addressed my prior comments

Reviewer #2 (Remarks to the Author):

Considering the queries raised authors choose to remove Hi-C data from results. Excluding it, authors have addressed most of the concerns convincingly supported by additional analysis and experiments. Few additional comments related to new analysis and other related queries should be clearly explained.

1. Authors at very high level show no strong difference between smokers and non-smokers groups by comparing variance of methylation levels between samples from. Along with this, it would be more meaningful to show if any difference exists on comparing levels of methylation for identified differential methylated probes/regions.
2. This is in relation to comment 8. Authors should show through distribution (boxplot) if there is significant difference in levels of methylation on comparing ESCC and ESAD.
3. Authors argue of using ANOVA due to no technical variations in experimental conditions. Can they compare these results with well-established linear models (such as limma) or counts (DESeq/edgeR) based analysis and show there are no significant differences?
4. In identifying differential methylated probes, did fold change was taken into consideration? If not, how the results would impact on considering 2 fold-change difference as threshold?
5. Levels of significance for association between lncRNA and hypo-methylated probes have now been provided. Can authors provide the same with hyper-methylated probes.
6. In YY1 knock-down experiments, along with ESCCAL-1 few other genes seems to be strong regulated. Does the pathway analysis of regulated genes match that with pathway on knock-down of ESCCAL-1?

Reviewer #4 (Replacement Reviewer to comment on Reviewer #3, Remarks to the Author):

In this manuscript, Cao et al. used an integrative-omics approach (WGS, WGBS, RNA-seq, and proteomic data) to delineate the epigenetic regulation of gene expression and identify oncogenic drivers of ESCC. However, the small sample size (10 pairs) and the lack of paired WGS data (only 3 matched patients have WGS data) make their results less convincing. Overall the integrative-omics analysis is not in-depth, and the main findings lack novel biological insights.

Major concerns:

1. The integrative-omics analysis is not in-depth. Although the authors claimed that they performed a multi-omics analysis (WGS, WGBS, RNA-seq and proteomic data), the main findings of this paper still focus on the relationship of promoter methylation (WGBS) and coding gene/lncRNA expression (RNAseq), which has been extensively studied before. There are not many novel biological insights from this paper.
2. Whole-genome bisulfite sequencing (WGBS) data in this study is one of the most valuable parts. However, the main findings of this paper only focus on promoter methylation. I think the 450K methylation array data (TCGA ESCC 450K dataset) can also find the same results. The authors need to do a more in-depth analysis with their WGBS data and add some novel and unique findings in their manuscript. I know that the authors have done some very primary analysis as a request by Reviewer #2, but that's not sufficient either.
3. The authors state strong enrichment of lncRNA in DMCs. However, lncRNAs are tissue-specific expression. Download all lncRNAs from GENCODE and do DMC enrichment does not make much sense.

Minor concerns:

1. The authors need to provide GEO dataset ID for raw datasets of WGS, WGBS, and RNA-seq in this study.

Authors' Response to Reviewer Comments

We appreciate the interest in our study and the thoughtful feedback. Below, we address each of the Reviewer's comments and add new data including in vivo studies to the revised manuscript. As a result of these changes, we believe the manuscript is improved. We hope the revised manuscript will now be suitable for publication. Additionally, several new analyses are provided for the Reviewers only for the sake of clarity and brevity in the revised manuscript. We are happy to include any of these at the suggestion of the Reviewers or Editors. Altogether, the new analyses do not alter the main findings and conclusions of our study.

Reviewers' comments:

Reviewer #1 (Remarks to the Author):

I have no further concerns. the authors have carefully addressed my prior comments

Response: We thank Reviewer #1 for the previous comments and constructive suggestions. We are doing our best to meet all of the Reviewers' expectations and to achieve a high-quality study.

Reviewer #2 (Remarks to the Author):

Considering the queries raised authors choose to remove Hi-C data from results. Excluding it, authors have addressed most of the concerns convincingly supported by additional analysis and experiments. Few additional comments related to new analysis and other related queries should be clearly explained.

1. Authors at very high level show no strong difference between smokers and non-smokers groups by comparing variance of methylation levels between samples from. Along with this, it would be more meaningful to show if any difference exists on comparing levels of methylation for identified differential methylated probes/regions.

Response: We thank the Reviewer for the suggestion. Based on the Reviewer's previous comments and the data included in our manuscript regarding alcohol but not tobacco exposure, we believe the Reviewer meant to state "alcohol consumers and non-alcohol consumers" and not "smokers and non-smokers". In the TCGA ESCC cohort (n=97), 70 patients have an alcohol consumption history, 25 are without an alcohol consumption history, and 2 samples are unannotated for alcohol consumption. We used 383,874 probes

Figure 1. Volcano plot of differentially methylated probes between alcohol-user group and non-alcohol-user group in TCGA-ESCC HM450K dataset. Each dot represents individual probes in HM450K array. Criteria for significantly differentially methylated probes is $|\log_2(\text{fold-change})| > 0.2$ AND adjusted p.value < 0.05 . The red dot = significantly hypermethylated probe; the blue dots = significantly hypomethylated probes.

for comparison between alcohol-user and non-alcohol-user groups, utilizing the limma R package, and this resulted in only three (3) significantly differentially methylated probes with the criteria of $|\log_2(\text{fold change})| > 0.2$ AND adjusted p.value < 0.05 (as shown in Figure 1). Now this analysis is included in the revised manuscript (Supplementary Figure 7c).

The three significant CpGs are in close proximity to three genes that are listed in Table 1 below. The CpGs probe (cg06834912) falls within the regulatory region of the FOXF1 gene and a literature search suggested that silent FOXF1 expression has a tumor suppressor function in breast cancer via promoter hyper-methylation (Cancer Res. 2010 Jul 15;70(14):6047-58). However, its association with alcohol consumption and functional roles in ESCC remain to be investigated. The other two probes are associated with non-coding RNAs whose functions are unknown to date.

Table 1. Differentially methylated probes between alcohol-users and non-alcohol-users in TCGA-ESCC HM450K dataset.

	Gene	chromosome	start	end	logFC	P.Value	adj.P.Val	Alcohol-users_vs_non-alcohol-users
cg06834912	FOXF1	chr16	86547202	86547204	0.27248973	3.87E-07	0.00032008	hypermethylation
cg02539153	PCDHB18	chr5	140614764	140614766	-0.2065779	0.00024119	0.0171999	hypomethylation
cg02159489	DQ585569	chr17	79459562	79459564	-0.3076041	0.00059008	0.03069899	hypomethylation

2. This is in relation to comment 8. Authors should show through distribution (boxplot) if there is significant difference in levels of methylation on comparing ESCC and ESAD.

Response: We thank the Reviewer for this suggestion. We have used TCGA-ESCA HM450K methylation data for orthogonal verification of DMCs and showed all methylation CpG probes could discriminate esophageal squamous carcinoma (ESCC) from esophageal adenocarcinoma (EAC/ESAD) samples. Among TCGA-ESCA HM450K methylation data, there are 97 ESCC and 89 EAC/ESAD samples. After the data were processed, there are a total of 381,132 CpGs probes with normalized β value in each sample. We used a linear model limma package for analysis of differentially methylated CpGs (DMCs) (Figure 2A). The volcano plot displays the significant differentially methylated CpGs in ESCC with colored dots, each dot represents a CpG probe (Figure 2B). Red dots indicate hyper-DMCs (N=3), blue dots indicate hypo-DMCs (N=24).

Based on the probe-Gene mapping

(https://tcga.xenahubs.net/download/probeMap/illuminaMethyl450_hg19_GPL16304_TCGA_legacy), we link 27 DMC associated-probes with nearby genes and 4 genes have multiple related CpG probes (Figure 3). These DMCs are located in CpG islands or shores, distributed among the genomic regions of promoters, introns and exons. Now this heatmap is included in the revised manuscript (Supplementary Figure 6c).

Figure 3. Hierarchical clustering of DMCs in ESCC relative to EAC. The CpG probes are listed on right side and their associated genes are listed on left side of heatmap. Samples in the column are clustered as either ESCC (green bar) or EAC (blue bar). hyper-DMCs= red color; hypo-DMCs = blue color.

Finally, we quantitatively show 6 DMC probe intensities in ESCC and EAC samples (Figure 4). Three hyper-DMC probes in ESCC relative to EAC: cg00172603 (chr1: 54822030-54822032) is located at intron 4 of the gene *SSBP3* (single-stranded DNA binding protein 3), whose function is as a regulator of mouse ESCs to differentiate into trophoblast-like cells (Liu et al. Stem Cell Res Ther. 2016; 7(1): 79. PMID: 27236334); cg01263942 (chr10: 695858-695860) is located at intron 1 of the gene *DIP2C* (disco-interacting protein 2 homolog C); loss of *DIP2C* triggers substantial DNA methylation and gene expression

changes, cellular senescence and epithelial-mesenchymal transition in cancer cells (Larsson et al. BMC Cancer. 2017;17(1):487. PMID: 28716088); cg14496282 is located in intron 4 of the gene *PDGFA* (platelet derived growth factor subunit A). The selected three hypo-DMC probes in ESCC versus EAC are: cg20691436 (chr10:5567477-5567479) is located at exon 1 of the gene *CALML3* (calmodulin-like 3); cg24884444 (chr8: 49572043-49572045) is located at intron 2 of the gene *EFCABI* (EF-hand calcium binding domain 1, non-coding RNA); cg25521254 (chr17:693136-693138) is located in intron 3 of the gene *RNMTL1* (RNA methyltransferase like 1). These are potential epigenetic biomarkers to distinguish the subtype of ESCC from EAC. Experimental verification is needed in a future study.

3. Authors argue of using ANOVA due to no technical variations in experimental conditions. Can they compare these results with well-established linear models (such as limma) or counts (DESeq/edgeR) based analysis and show there are no significant differences?

Response: We thank the Reviewer for the comment. As we have described before, there are two main platforms for DNA methylation profiling: methylation array and high-throughput sequencing. For differential methylation, both parametric and non-parametric statistical tests are used directly on the beta value (methylation ratio of methylated versus total signal intensity at a CpG site) [Dedeurwaerder et al. 2014;15:929-941]. Whereas methylation sequencing data analysis are typically analyzed by 2 approaches: (1) ratio-based algorithms and (2) count-based model algorithms. We learned in a study conducted by Zhang et al (cited

in Figure 5 legend) that ratio-based methods outperform count-based methods in terms of type I error rate control including ANOVA (Figure 5).

Zhang, et al., Statistical method evaluation for differentially methylated CpGs in base resolution next-generation DNA sequencing data, *Briefings in Bioinformatics*, Volume 19, Issue 3, May 2018, Pages 374–386, by permission of Oxford University Press

Figure 5. Observed type I error rates under the null hypothesis. The left panel is for ratio-based, and the right is for count-based statistical models. The numbers on the bars are the observed type I error rates. The dashed line is expected type I error rate ($\alpha=0.05$). Cited from Zhang Y. et al. Statistical method evaluation for differentially methylated CpGs in base resolution next-generation DNA sequencing data. *Briefings in Bioinformatics*. 2018 May 1;19(3):374-386. doi: 10.1093/bib/bbw133. PMID: 28040747

As the Reviewer suggested, we applied a ratio-based approach to obtain 18,471,772 CpGs in each normal and tumor sample. The limma linear model was used to define differentially methylated CpGs (DMCs) between ESCC and normal samples. As shown in Figure 6, with the criteria of $FDR \leq 0.05$, Limma analysis resulted in 5,768,882 DMCs. When compared to our previous ANOVA analysis (Figure 6), there are 5,039,274 DMCs common to both analyses. Although ANOVA and Limma show 53,571 and 72,968 unique DMC callings, respectively, there is a no ground truth tool to measure the accuracy of each method. Therefore, we could not justify the use of one method over the other and we have left the analysis unchanged in the revised manuscript as a result.

Figure 6. Venn diagram comparison of ANOVA and Limma algorithms to define the differentially methylated DMCs in ESCC. Hypergeometric test indicates a significant overlap of DMCs between the two statistical methods ($p < 0.001$).

4. In identifying differential methylated probes, did fold change was taken into consideration? If not, how the results would impact on considering 2 fold-change difference as threshold?

Response: We thank the Reviewer for the comment. The fold-change was considered in our initial exploratory analysis (Figure 8). We started with 18,421,444 CpGs across all samples (normal and tumors) following quality control. Each CpG has a methylation level (methylation level = methylated reads/total reads at a given CpG). After comparing the means of the methylation levels in tumors with the means of the corresponding methylation levels in normal samples using the ANOVA approach and adjusted p -value ≤ 0.05 , we obtained 5,092,845 differentially methylated CpG (DMCs). If we applied fold-change, for example, a 2-fold for hyper and $\frac{1}{2}$ -fold for hypo in tumor, respectively, as highlighted with two green lines as the fold-change boundary, we would lose 40% of significant differentially methylated CpGs DMCs in the green area. This stringent cut-off could give rise to a higher false negative rate, which we elected to avoid in our initial exploratory and screening analysis.

We also realized that CpG methylation change is not as effective as differential gene-expression analysis because gene expression levels vary much more widely with the abundance of gene expression, ranging from zero to hundreds of thousands, whereas the range of DNA methylation levels is normalized between 0 - 1. Additionally, a fold-change can distort information. For example, if a CpG methylation is 0.6 in a tumor sample, while the same CpG methylation is 0 in a matched normal sample, the ratio = $0.6/0$, which is infinity.

5. Levels of significance for association between lncRNA and hypo-methylated probes have now been provided. Can authors provide the same with hyper-methylated probes.

Response: We thank the Reviewer for pointing this out. We used 2105 lncRNAs, which have an overlap with hypermethylated DMCs, for the permutation test. It turns out that these lncRNAs are also significantly enriched in hypermethylated DMCs when compared to random genomic regions (Figure 8). Although twice as many lncRNAs are located in hypomethylated DMCs, when they are compared with random genomic regions using a permutation test, lncRNAs do not show an enrichment preference for either hyper- or hypomethylated DMCs. This leads us to reject the alternative hypothesis that lncRNAs are enriched in hypomethylated DMCs. Therefore, we decided to not show this result in main Figure 6a from the revised manuscript.

6. In YY1 knock-down experiments, along with ESCCAL-1 few other genes seems to be strong regulated. Does the pathway analysis of regulated genes match that with pathway on knock-down of ESCCAL-1?

Response: We thank the Reviewer for the comment. We have conducted this analysis before and found the gene sets overlapping between siYY1 and shESCCAL-1 are not significant. We have performed this analysis in two different ways. First, we performed Gene Set Enrichment Analysis (GSEA, Broad Institute) in the global RNA-seq datasets, where we queried for differential gene sets between siCtrl (n=2) and siYY1 (n=2) cohorts. Unlike in our previous analysis of shESCCAL-1 cohort, where protein synthesis related genes were identified differentially regulated (Supplementary Figure 30a), transmembrane receptor mediated signaling and golgi vesicular-trafficking related gene sets were mostly differentially upregulated or downregulated, respectively (Figure 9a).

Second, we also defined the top downregulated genes in the two replicates of the siYY1 samples (siYY1 gene set), compared to the siCtrl samples (Fold change ≥ 2 , FDR < 0.05). We then performed a hypergeometric test between the siYY1 gene set (n=125) and previously described a 'shESCCAL-1 gene set' (n=561). The p-value for the hypergeometric test was 0.5 and, hence, it was deemed insignificant (Figure 9b).

Also, we found in our analysis that ESCCAL-1 was ranked 19th within the most differentially downregulated gene upon YY1 knock down. From these observations, we concluded that YY1 regulates many different downstream biological components and not just ESCCAL-1 exclusively. Interestingly, YY1 knockdown also differentially perturbed expression of a confluence of other long non-coding RNAs [e.g. LOC101929613(rank 2nd), LOC101927418 (rank 4th), LOC101930071(rank 5th), LOC103021295 (rank 6th) and LOC102724238 (rank 7th)]; nevertheless, the biology for these changes remains unclear and it is worth investigating further as a separate research endeavor.

Reviewer #4 (Replacement Reviewer to comment on Reviewer #3, Remarks to the Author):

In this manuscript, Cao et al. used an integrative-omics approach (WGS, WGBS, RNA-seq, and proteomic data) to delineate the epigenetic regulation of gene expression and identify oncogenic drivers of ESCC. However, the small sample size (10 pairs) and the lack of paired WGS data (only 3 matched patients have WGS data) make their results less convincing. Overall the integrative-omics analysis is not in-depth, and the main findings lack novel biological insights.

Response: We appreciate the Reviewer's concerns. We have addressed the sample size issues in our previous response and acknowledged this caveat in the revised manuscript. We continue to emphasize that our comprehensive analyses and our experimental verification of the exploratory findings is the largest WGBS in any single type of cancer (ESCC). Along with

additional, orthogonal TCGA-esophageal cancer data and our own independent samples (~100 paired ESCC samples), we are developing an understanding of the genome-wide non-canonical epigenetic regulation of gene expression and how methylome alterations affect regulatory regions and non-coding RNA expression (e.g., long non-coding RNA, ESCCAL-1) in ESCC.

Major concerns:

1. The integrative-omics analysis is not in-depth. Although the authors claimed that they performed a multi-omics analysis (WGS, WGBS, RNA-seq and proteomic data), the main findings of this paper still focus on the relationship of promoter methylation (WGBS) and coding gene/lncRNA expression (RNAseq), which has been extensively studied before. There are not many novel biological insights from this paper.

Response: We appreciate the Reviewer's concerns. As seen in our previous response, we have extended the analyses on methylation aberrations in different regions in the genome such as promoter regions, enhancers, gene bodies and CpG island regions, DNA accessibility and many transcription-factor binding regions. These data provided the evidence to explain potential mechanisms of non-canonical epigenetic regulation of gene expression, and we experimentally examined higher promoter methylation and high expression of WNT2 as one illustrative example in this study. We now provide in vivo tumor xenograft data demonstrating that WNT2 functions as a driver in ESCC (Figure 10). To our knowledge, this has never before been reported. Therefore, we believe our findings do contribute to the understanding of complexity and functionality of gene regulatory networks in the development and progression of ESCC.

2. Whole-genome bisulfite sequencing (WGBS) data in this study is one of the most valuable parts. However, the main findings of this paper only focus on promoter methylation. I think the 450K methylation array data (TCGA ESCC 450K dataset) can also find the same results. The authors need to do a more in-depth analysis with their WGBS data and add some novel and unique findings in their manuscript. I know that the authors have done some very primary analysis as a request by Reviewer #2, but that's not sufficient either.

Response: We appreciate the Reviewer's concern. Based on previous Reviewers' suggestions, we have added the impacts of 5 million DMCs on ESCC-related enhancers, genome-wide CpG islands regions, and the interaction of enhancers and promoters that cooperatively regulate downstream gene expression. We also added the differential effects of transcription factor finding sites on enhancer regions and promoter regions. All of these systematic analyses support the diverse epigenetic regulation of gene expression in ESCC. This is new

and provides novel contributions to the field. We will incorporate your concern in our next investigations and their subsequent publication.

3. The authors state strong enrichment of lncRNA in DMCs. However, lncRNAs are tissue-specific expression. Download all lncRNAs from GENCODE and do DMC enrichment does not make much sense.

Response: We thank the Reviewer for the comment. From our WGBS analysis, we found 5 million, differentially methylated cytosines (DMCs) from the ESCC genome relative to the normal esophagus epithelium genome, and 98% of them are a loss of methylation. In further annotation of the hypo-DMCs, 43.23% of them are located at poorly annotated regions of the genome. The question we asked logically is whether lncRNA genes overlap the hypo-DMCs, and to what degree. Nevertheless, we agree that this is a more descriptive analysis, but if the promoter regions of lncRNAs are hypomethylated, we can infer that there is a biological consequence of such as high expression in cancer. This is supported with the current observation that recurrent hypomethylation of lncRNAs (1,006 out of 6,475) in 33 types of cancers is associated with high levels of the transcripts in cancers (Wang et al. Cancer cell. 2018; 33:706-720. PMID: 29622465). Our data identified only several lncRNAs whose expression levels are anti-correlated with promoter methylation alterations (main Figure 6b). Therefore, when we put this analysis in context, it begins to make biological sense.

Minor concerns:

1. The authors need to provide GEO dataset ID for raw datasets of WGS, WGBS, and RNA-seq in this study.

Response: We will provide the data access ID for all the datasets in the manuscript.

REVIEWERS' COMMENTS:

Reviewer #2 (Remarks to the Author):

Authors have addressed all comments raised with relevant details and other supporting data. Data related to one of comments have now been removed from the manuscript as it is proven to be not significant.

Reviewer #3 (Remarks to the Author):

The authors have adequately addressed my concerns.

REVIEWERS' COMMENTS:

Reviewer #2 (Remarks to the Author):

Authors have addressed all comments raised with relevant details and other supporting data. Data related to one of comments have now been removed from the manuscript as it is proven to be not significant.

Response: *We thank Reviewer #2 for the previous comments and constructive suggestions. We are doing our best to meet all of the Reviewers' expectations and to achieve a high-quality study.*

Reviewer #3 (Remarks to the Author):

The authors have adequately addressed my concerns.

Response: *We thank Reviewer #3 for the satisfaction with our scientific responses. We are doing our best to meet all of the Reviewers' expectations and achieve a high-quality study.*